# Low variability of the Atlantic Meridional Overturning Circulation throughout the Holocene

Lukas Gerber [1] ✉, Jörg Lippold [1], Finn Süfke[1], Ole Valk [1], Pierre Testorf[2,3], Manuel Ehnis[1], Saskia Tautenhahn[1], Lars Max[4], Cristiano M. Chiessi [5], Marcel Regelous[6], Sönke Szidat [2,7], Oliver Friedrich [1] & Frerk Pöppelmeier [2,3]

Earth system models and paleo-reconstructions indicate that shifts in Atlantic Meridional Overturning Circulation (AMOC) strength profoundly impact global climate. While the last glacial termination experienced large AMOC variations, evidence of AMOC changes during the Holocene are poorly constrained. Here we present a Holocene AMOC reconstruction by quantifying mean bottom water advection strength in the deep North Atlantic. For this, we estimated volumetric flow rates from sedimentary $^{231}Pa/^{230}Th$ records with millennial resolution using the Bern3D model. We found that while during the Early Holocene the AMOC recovered from its weak deglacial state, it experienced a weakening between 9.2 to 8 ka BP, coinciding with North Atlantic meltwater pulses. From 6.5 ka BP onward, the AMOC strength stabilized, reaching its pre-industrial state around ~18 Sv. Hence, according to future projections, anthropogenic climate change may result in an AMOC slowdown unprecedented for most of the ongoing Holocene interglacial.

The Atlantic Meridional Overturning Circulation (AMOC) is a crucial component of Earth's climate system, influencing regional to global climate change. Variations in its strength drive anti-phased temperature anomalies between the Northern and Southern Hemispheres as a consequence of the thermal bipolar seesaw[1]. Reconstructions of past climate and ocean circulation show evidence of these changes in the form of rapid shifts in Greenland temperatures[2] as well as sea-surface temperatures in the North Atlantic[3]. These shifts, termed Heinrich and Dansgaard-Oeschger Events, punctuated the climate of the last glacial period. This behaviour is assumed to be a result of the inherent multistability of this important tipping element in the Earth system[4].

Reconstructing past variations of the AMOC is thus critical for the assessment of its variability on timescales extending beyond modern observations, which only span a few decades[5]. These reconstructions further provide crucial information for disentangling the anthropogenic response from natural variability, and for the ongoing debate whether the AMOC may have already weakened during recent decades[6–8]. Consequently, the question arises whether AMOC variability might have contributed to climate changes during the Holocene (11,700 years before present (BP) to today), that may serve as analogues for the future. There is a general consensus that the variability of the AMOC was subdued during the current interglacial period[9–11] compared to the last glacial termination[12]. In particular, protactinium-231 to thorium-230 ($^{231}Pa/^{230}Th$) reconstructions imply large swings in the AMOC's strength during the glacial and deglacial[13,14]. Therefore, long-term constraints are urgently needed to better characterize the natural state and variability of the AMOC over the Holocene. For this purpose, various proxies have been utilized to reconstruct past AMOC variability; most of these approaches, however, have been limited to mainly qualitative assessments[11,15–22].

[1]Institute of Earth Sciences, Heidelberg University, Heidelberg, Germany. [2]Oeschger Centre for Climate Change Research, University of Bern, Bern, Switzerland. [3]Climate and Environmental Physics, Physics Institute, University of Bern, Bern, Switzerland. [4]MARUM Center for Marine Environmental Sciences, University of Bremen, Bremen, Germany. [5]School of Arts, Sciences and Humanities, University of São Paulo, São Paulo, Brazil. [6]GeoZentrum Nordbayern, FAU Erlangen-Nürnberg, Erlangen, Germany. [7]Department of Chemistry, Biochemistry and Pharmaceutical Sciences, University of Bern, Bern, Switzerland. ✉e-mail: lukas.gerber@geow.uni-heidelberg.de

Here, we employ the ratio of $^{231}$Pa/$^{230}$Th, measured from deep-sea sediments, to infer past strengths of the AMOC during the Holocene. $^{231}$Pa and $^{230}$Th are both radioactive decay products of $^{235}$U and $^{238}$U, respectively, which are well-mixed in ocean water, resulting in the assumption of a spatiotemporally constant production rate of 0.093[23]. Bound to particles, $^{231}$Pa and $^{230}$Th are removed from the ocean and are eventually buried in bottom sediments.

The $^{231}$Pa/$^{230}$Th proxy is based on the differing removal rates, with the more particle-reactive $^{230}$Th being adsorbed more intensely than $^{231}$Pa. As such, $^{231}$Pa exhibits a longer residence time in the ocean compared to $^{230}$Th and can be transported away from its production site along with advecting water masses. In the modern ocean, 26% of the produced Pa is exported out of the North Atlantic along with North Atlantic Deep Water (NADW)[24–26]. This results in sedimentary $^{231}$Pa/$^{230}$Th ratios significantly below the production ratio of 0.093 in the majority of the North Atlantic[25]. This simplified conceptual view thus suggests that during times of weaker AMOC less $^{231}$Pa would be advected southwards and a sediment core in the North Atlantic would hence record higher $^{231}$Pa/$^{230}$Th values as inferred from model approaches of different complexities[27–31].

The $^{231}$Pa/$^{230}$Th proxy has been widely applied to the last glacial and deglaciation periods, but for the Holocene, two high-resolution records have only recently provided evidence for a rather stable AMOC during that time[9,10]. A specific focus of both studies was a close investigation of the AMOC at around 8 ka BP, for which ice-core records from Greenland (NGRIP δ$^{18}$O) indicate a centennial-scale cold spell[2]. This is commonly known as the 8.2 ka BP cooling event, that has also been identified throughout a wide array of other climate archives[32,33]. For the 8.2 ka BP event, Earth system models have simulated a global climate impact by weakening the AMOC which curtails the northward oceanic heat transport inducing various changes in the atmospheric circulation[34,35]. These high-resolution $^{231}$Pa/$^{230}$Th reconstructions, however, found no significant deviation from the Holocene baseline for this event[9,10]. Both previously published $^{231}$Pa/$^{230}$Th records of Holocene AMOC variability show a relatively constant baseline, not following the Northern Hemisphere temperature trend of continuous cooling during the Late Holocene[36], which would imply a steady decrease in AMOC strength after the Holocene Thermal Maximum around 7 ka BP[37,38]. However, as both published records were retrieved from locations in relative proximity to each other in the sub-tropical western North Atlantic, it remains unclear how well they represent the overall AMOC strength.

In this study, we present two new $^{231}$Pa/$^{230}$Th records (ODP Site 983 and GeoB18529-2) and increased the temporal resolution of two previously published $^{231}$Pa/$^{230}$Th records (ODP Site 1063 and ODP Site 1059)[9,10], achieving millennial-scale resolution (Supplementary Table 1) across a wide range of locations in the North Atlantic (Fig. 1). By employing the $^{231}$Pa/$^{230}$Th-enabled Bern3D Earth system model, we further estimate from these proxy records the Holocene mean AMOC strength, thus quantifying past AMOC changes.

## Results and discussion
### Down-core $^{231}$Pa/$^{230}$Th profiles
Here, we present two new millennial-scale resolution $^{231}$Pa/$^{230}$Th down-core records covering the entire Holocene. We also added new $^{231}$Pa/$^{230}$Th data points to two previously published $^{231}$Pa/$^{230}$Th records[10,39], to archive a similar millennial-scale temporal resolution (Fig. 2). All records exhibit different $^{231}$Pa/$^{230}$Th baselines, which do not indicate conflicting results with regard to the Holocene AMOC strength, but are the result of the proxy behaviour, as also observed in the modern North Atlantic with a strong dependency on location and water depth[40]. These location-dependant differences are a function of regional particle fluxes, the geometry of the NADW overturning cell (defined as the distance to the deep-water formation regions), and the circulation path. As suggested by model simulations[29,41], our shallower

cores (ODP 983 and KN140-2-51GGC) generally exhibit higher $^{231}$Pa/$^{230}$Th values than deeper cores (ODP 1063 and ODP 1059), reflecting the growing influence of deep $^{231}$Pa export by the AMOC. This relation between water depth and $^{231}$Pa/$^{230}$Th applies to the two northernmost cores (ODP 983 and GeoB18529-2), albeit both having on average higher absolute $^{231}$Pa/$^{230}$Th ratios compared to the other cores as a result of being more proximal to the regions of deep-water formation and as a result of higher particle concentration in these northern regions. It is thus expected that past changes in the strength and the overturning geometry of NADW led to different responses in $^{231}$Pa/$^{230}$Th at different locations. Accordingly, it is advisable to interpret the variability of $^{231}$Pa/$^{230}$Th within one individual down-core $^{231}$Pa/$^{230}$Th profile in terms of AMOC variability, rather than interpreting absolute differences between different core locations[42].

The $^{231}$Pa/$^{230}$Th values of all records show a decreasing trend from the beginning of the Holocene (11.7 ka BP) to about 10 ka BP. A notable exception is ODP 983 (Fig. 2a), which exhibits an inverted behaviour compared to the trends evident in the other five cores, as predicted by recent model simulations for this location and water depth under a post-deglacial strengthening of the AMOC[43,44]. From 10 ka BP onward, the $^{231}$Pa/$^{230}$Th records stayed largely stable for the remainder of the Holocene with only two evident minor excursions. Compared to the time periods directly before and after, the interval from 9 to 8 ka BP shows slightly elevated $^{231}$Pa/$^{230}$Th ratios at the study sites but ODP 983 (Fig. 2b–d). The second feature is only apparent at ODP 1063: between 4.3 and 3.8 ka BP the $^{231}$Pa/$^{230}$Th ratios are also slightly elevated (Fig. 2c). Yet, this excursion is not recorded in the other data sets (Fig. 2).

To assess millennial-scale variability, we detrended the $^{231}$Pa/$^{230}$Th records by employing a generalized additive model (GAM) fit. For the GAM, uniform and low smoothing parameters were used, to exclusively remove prolonged and linear trends from the different $^{231}$Pa/$^{230}$Th profiles. Subsequently, the linear trends extracted by the GAM were subtracted from the individual $^{231}$Pa/$^{230}$Th profiles, enabling the calculation of standard deviation (and variance) for each individual profile (Table 1). Overall, a low variability within the examined time period for all five high-resolution $^{231}$Pa/$^{230}$Th records is observed (RSD < 10%), whereby the RSD of 8.4% for GeoB18529-2 is the highest RSD.

A close temporal co-variation between bOpal and $^{231}$Pa/$^{230}$Th would imply that the observed variability of $^{231}$Pa/$^{230}$Th was controlled by bOpal. Thus, it is important to cross-check the $^{231}$Pa/$^{230}$Th results of each individual core with its respective bOpal record (Supplementary Fig. 1) in order to estimate a possible bOpal influence on the $^{231}$Pa/$^{230}$Th records. The measured bOpal concentrations range from below 1 to about 7 wt% for all cores during the entire Holocene. Similarly to $^{231}$Pa/$^{230}$Th, the down-core bOpal concentrations show only limited temporal variations as well and no significant or coherent trend during the investigated time interval (see Supplement and Supplementary Fig. 2).

### Holocene AMOC strength
The compiled $^{231}$Pa/$^{230}$Th records show relatively constant values throughout the Holocene, but with substantially different absolute values of $^{231}$Pa/$^{230}$Th and partially inverted long-term trends. These differences are mainly related to their respective locations within the overturning cell of NADW[31,44]. In addition to AMOC strength, particle type and concentration, the position of the core site significantly influences the sedimentary $^{231}$Pa/$^{230}$Th ratios. The water depth determines the extent of the $^{231}$Pa-deficit, caused by the advective export of $^{231}$Pa within NADW. The deeper the water depth, the higher the fraction of above produced $^{231}$Pa to be advected away relative to $^{230}$Th. This effect is observed at core-top sediments in a wide range of the Atlantic Ocean south of 40°N (i.e., south of the deep-water formation zones). For most of the Atlantic Ocean $^{231}$Pa/$^{230}$Th thus decreases with water depth[40,45,46]. This leads to the challenge that $^{231}$Pa/$^{230}$Th records from

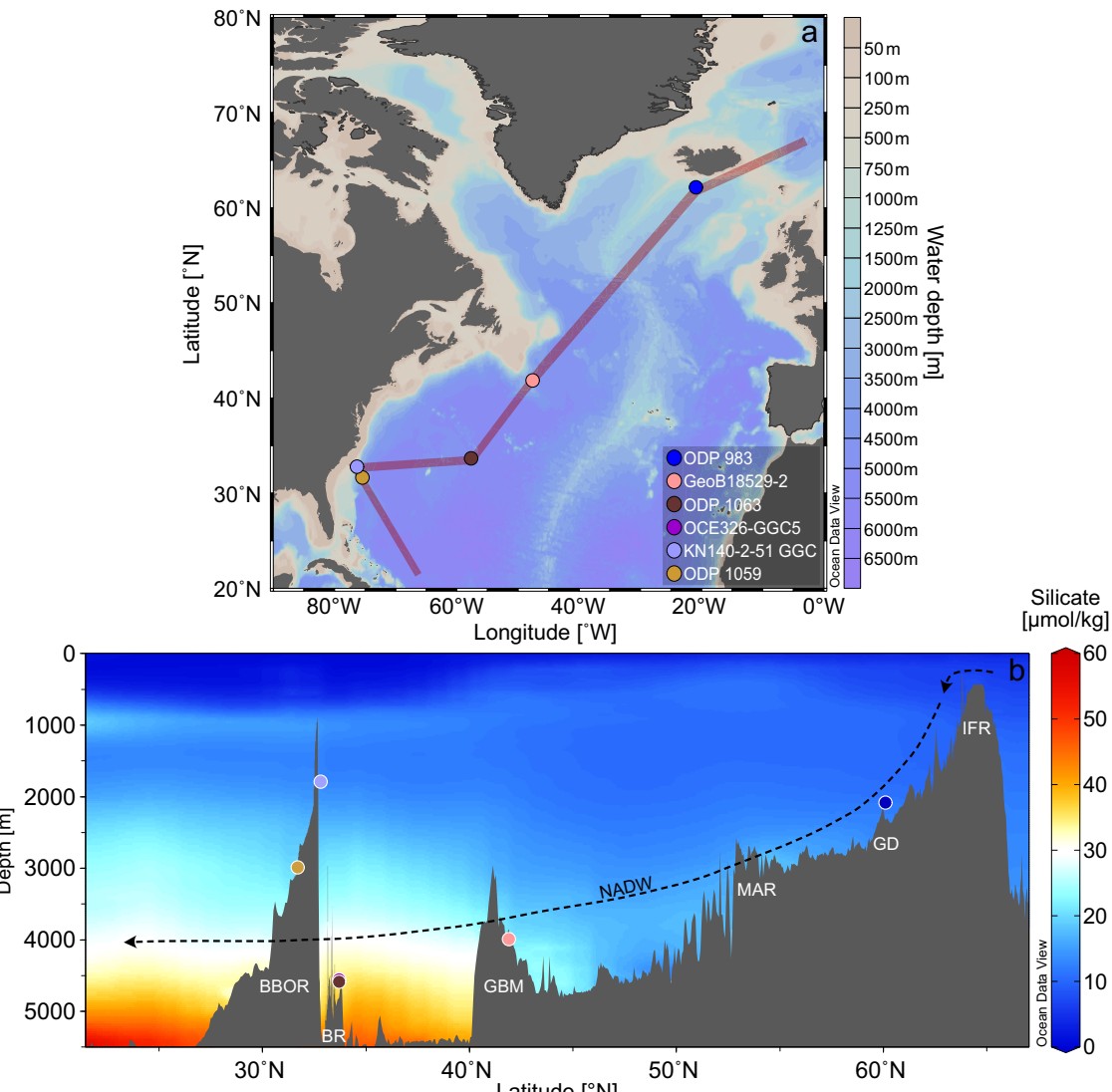

**Fig. 1 | Locations of investigated sites. a** Overview of site locations, connected by a transect through the North Atlantic (red line). **b** Modern silicate content[100] of the North Atlantic, along the transect depicted in (**a**). The dashed arrow indicates the general flow direction of North Atlantic Deep Water (NADW). Site locations are indicated as filled circles. New $^{231}$Pa/$^{230}$Th records: ODP 983 and GeoB18529-2. $^{231}$Pa/$^{230}$Th records with new additional data: ODP 1063 and ODP 1059. Unchanged previously published $^{231}$Pa/$^{230}$Th records: OCE326-GGC5 and KN140-2-51 GGC. BBOR = Blake Bahamas Outer Ridge; BR = Bermuda Rise; GBM = Grand Banks Margin; MAR = Mid Atlantic Ridge; GD = Gardar Drift; IFR = Iceland-Faroe-Ridge. This figure was partially produced with Ocean Data View[101].

different locations are difficult to compare in the sense of reflecting the general AMOC strength and even more difficult to convert into a quantitative strength of the past AMOC. To resolve this issue, we combined the geochemical results of this study with results of the proxy-enabled Bern3D model[47]. The Bern3D model includes an explicit implementation of the $^{231}$Pa/$^{230}$Th proxy, such that sedimentary $^{231}$Pa/$^{230}$Th at individual core locations can be directly associated with changes in AMOC strength (see "Methods").

First, we assessed the detectability of simulated $^{231}$Pa/$^{230}$Th ratios to different AMOC perturbation scenarios at our examined core locations, by simulating 21 idealized AMOC slowdown scenarios with varying durations (100, 120, 140, 160, 180, 200 and 300 years) and amplitudes (−4.9, −6.8 and −8.5 Sv, weaker compared to the pre-industrial (PI) AMOC state). For the direct comparisons to the Holocene reconstructions, site-specific factors are considered including the respective sedimentation rate, water depth, sampling interval (Supplementary Table 1) and bioturbation[10] of each location. The simulated AMOC perturbations resulted in occasionally very different responses in $^{231}$Pa/$^{230}$Th at each core location (Supplementary Fig. 3). We assume

that an excursion in a site-specific timeseries significantly deviates from its background variability and is hence detectable if it exceeds its natural variability during the Holocene by two standard deviations (Table 1). These results indicate that the relatively shallow site KN140-2-51GGC (1790 m water depth) has a limited sensitivity for capturing AMOC variability, as its response in $^{231}$Pa/$^{230}$Th to transient AMOC changes is minimal and never exceeds our threshold for detectability (Fig. 3). For this reason, despite its high temporal resolution, the $^{231}$Pa/$^{230}$Th record of KN140-2-51GGC is not considered in the following comprehensive $^{231}$Pa/$^{230}$Th-based synopsis for the Holocene AMOC. These sensitivity simulations also show that −4.9 Sv AMOC weakening over a short (100–160 years) period cannot be detected by any core site. The smallest and shortest AMOC perturbation, that can be reliably detected is an AMOC weakening of −6.8 Sv for 180 years. Shorter or smaller AMOC excursions are difficult or impossible to detect due to analytical noise, bioturbation, and limited sampling resolution (Fig. 3).

Next, we evaluated the site-specific relationships between $^{231}$Pa/$^{230}$Th and AMOC strength under PI conditions, with 20 idealized simulations of slow transient changes in AMOC strength forced by

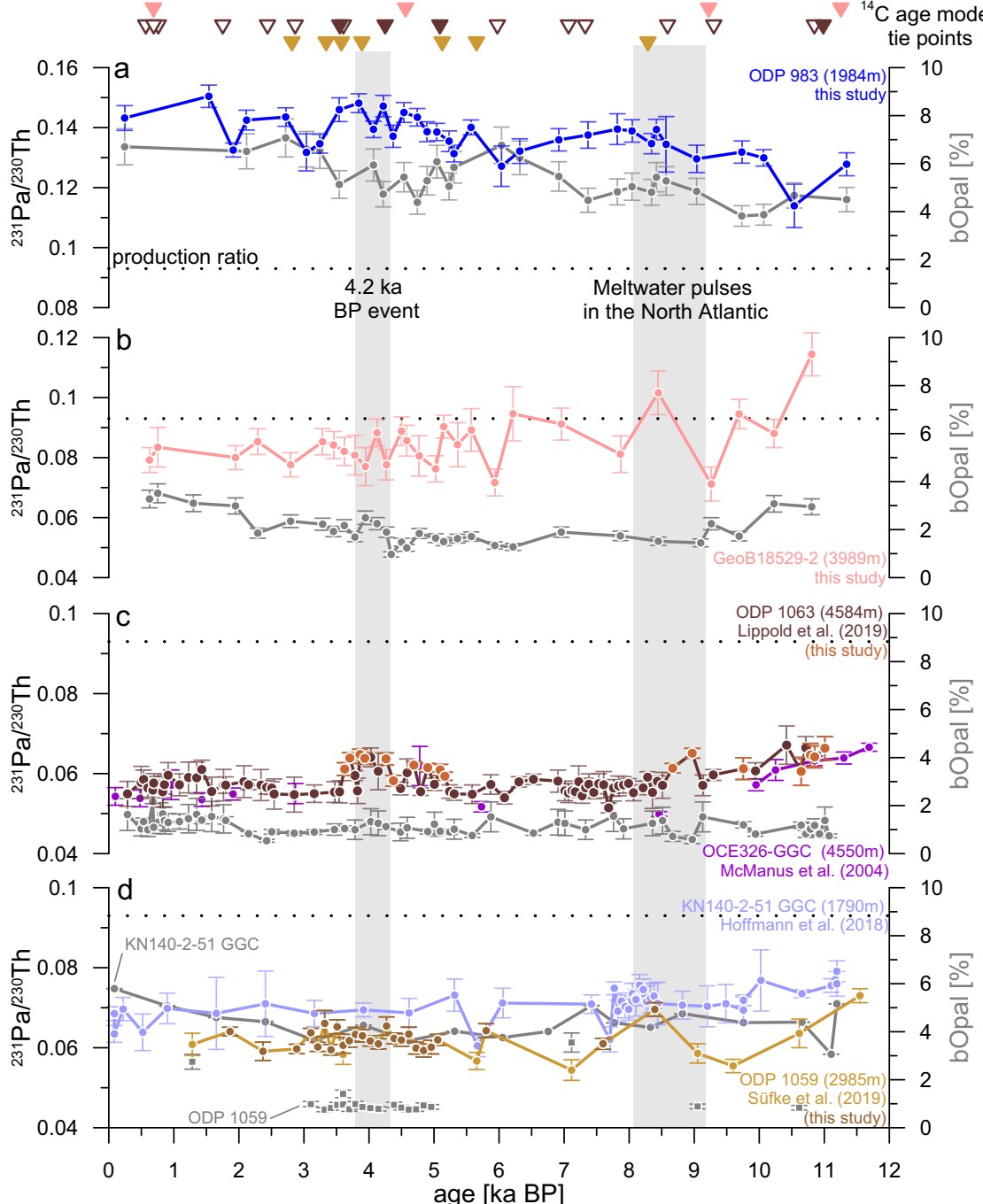

**Fig. 2 | Holocene $^{231}$Pa/$^{230}$Th (non-grey lines) and biogenic Opal (bOpal, grey lines) results from the six sediment cores investigated in this study.** The error bars indicate the standard errors associated with each data point. Sites (**a**) to (**d**) arranged according to their latitude in the North Atlantic (top = northernmost, bottom = southernmost)[9,10,24,81]. The triangles (colored accordingly to their respective $^{231}$Pa/$^{230}$Th records) at the top of this figure mark the tie points used for the newly generated or updated age models with filled triangles representing new $^{14}$C age tie-points of this study, and empty triangles symbolising already published $^{14}$C data. The vertical grey bars highlight the timing of multiple meltwater pulses in the North Atlantic[49] around 8.2 ka BP and the 4.2 ka BP event. The horizontal dotted black lines represent the $^{231}$Pa/$^{230}$Th production ratio of 0.093[25] in the ocean.

freshwater hosing to the North Atlantic (see "Methods"). These freshwater forcings were adjusted in a way that targeted AMOC strengths were achieved. With this approach, we obtained a range of AMOC strengths along with their corresponding pseudo-proxy $^{231}$Pa/$^{230}$Th ratios for each site analyzed in this study. These model simulations explain the opposing trends in the down-core profiles during the early Holocene between records from different core sites (e.g., ODP 983 versus ODP 1063). Whether $^{231}$Pa/$^{230}$Th records exhibit a positive or negative correlation with AMOC strength depends mainly on water depth and on the complex interactions between removal by particle concentrations and the net effect of export versus import by advection[43]. Although these effects are difficult to disentangle in a conceptual model due to inter-dependencies (e.g., particle concentrations depend on AMOC strength via its controlling factor on nutrient replenishment and export production) and far versus near-field effects, they are considered in the Bern3D Earth system model[47]. Based on these site-specific $^{231}$Pa/$^{230}$Th to AMOC relationships derived from the Bern3D model simulations (see "Methods"), the

**Table 1 | Results of the GAM (Generalized Additive Model) detrended (det.) $^{231}$Pa/$^{230}$Th profiles**

| Core ID | Mean $^{231}$Pa/$^{230}$Th (det.) | SD (det.) | RSD (det.) |
|---|---|---|---|
| ODP 983 | 0.137 | 0.005 | 3.7% |
| GeoB18529-2 | 0.085 | 0.007 | 8.4% |
| ODP 1063 | 0.058 | 0.003 | 5.5% |
| ODP 1059 | 0.062 | 0.003 | 5.0% |
| KN140-2-51 GGC | 0.071 | 0.003 | 4.4% |

The mean detrended $^{231}$Pa/$^{230}$Th values at each site, along with their associated detrended standard deviations (SD) and relative standard deviations (RSD) are reported.

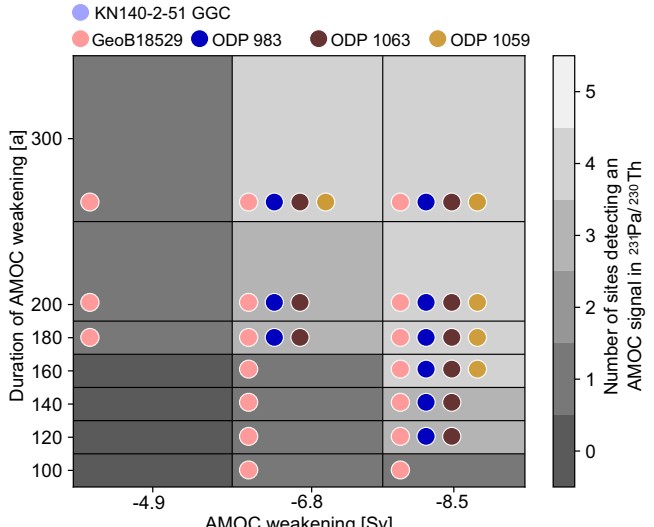

**Fig. 3 | Detectability of Atlantic Meridional Overturning Circulation (AMOC) perturbations.** Idealized scenarios with different durations of perturbations (100, 120, 140, 160, 180, 200 and 300 years) and magnitudes of potential AMOC weakening (-4.9, -6.8, −8.5 Sv, compared to pre-industrial) at the five core sites investigated in this study were tested to assess whether these sites exhibit sufficient (> 2σ) sensitivity to detect significant $^{231}$Pa/$^{230}$Th signals. Core KN140-2-51 GGC does not record significant signals in $^{231}$Pa/$^{230}$Th for neither of the above examined scenarios of AMOC weakening.

individual sedimentary $^{231}$Pa/$^{230}$Th data (Fig. 2) of all cores were collectively converted into an AMOC strength value, expressed in Sverdrup (Sv). The resulting Sv values were then smoothed using a GAM fit, producing a continuous timeseries of reconstructed mean AMOC strength in the North Atlantic during the Holocene (Fig. 4) with a sub-millennial temporal resolution (13.5 $^{231}$Pa/$^{230}$Th data points per 1ka).

**Early Holocene and 8.2 ka event**

The onset of the early Holocene is marked by a gradual recovery of AMOC strength from the deglacial perturbation associated with the Younger Dryas (YD), during which the AMOC was significantly weaker, resulting in cooling of the Northern Hemisphere[48]. Over the following 2000 years, the AMOC gradually intensified, reaching a state at around 9.5 ka BP that was similar to, but slightly weaker than PI strength. Starting at 9.2 ka BP, the AMOC experienced an abrupt weakening of up to 3 ( ± 1.3) Sv, relative to PI levels, that paused for approximately 1 ka the general trend of AMOC strengthening (Fig. 4b). This decline coincides with the final melting of the Laurentide Ice Sheet (LIS) and the associated freshwater influx into the North Atlantic from sources such as the drainage of Lake Agassiz, and changes in North American continental freshwater routing[49,50]. These meltwater fluxes are

evidenced in the increased rates of relative sea level rise during this period[51].

This period also encompasses the 8.2 ka event, representing the most significant cooling episode in Greenland ice core records over the last 12,000 years (Fig. 4a). It is hypothesized that a large meltwater outburst from North American proglacial lakes triggered an abrupt weakening of the AMOC, leading to reduced northward heat transport, followed by a subsequent AMOC recovery and Greenland temperature rebound[34]. Our reconstructed AMOC perturbation aligns well with the periods of increased sea level rise between 9-8 ka BP (Supplementary Fig. 4), suggesting that it could have been caused by either multiple individual meltwater pulses, an overall increased freshwater flux over this time period or a combination of both factors. While the 8.2 ka event is present in the δ$^{18}$O records of the NGRIP (Fig. 4a), our data do not resolve AMOC perturbations shorter than approximately 180 years (Fig. 3), a duration estimated for the 8.2 ka BP event. This discrepancy is likely due to the influence of sedimentary bioturbation on the $^{231}$Pa/$^{230}$Th record, smoothing the record and reducing the amplitude of short-term events (a duration of 100–130 years for the 8.2 ka event) (Fig. 3)[10,50,52]. As a result, our record primarily captures mean sub-millennial-scale AMOC variability in the western North Atlantic, effectively averaging out potential short-term AMOC slowdowns associated with individual meltwater pulses such as the 8.2 ka event[49]. Consequently, the $^{231}$Pa/$^{230}$Th-based reconstructed AMOC reduction (Fig. 4b) of around 15% relative to PI levels, is smaller compared to previous model studies[50,52]. However, recent modelling results suggest that the Holocene AMOC may have been less sensitive to freshwater fluxes than previously assumed, indicating that the weakening of the AMOC during the 8.2 ka event may have been less severe and indeed more subdued than previously thought, potentially supporting our results[53].

**Mid-Holocene and 4.2 ka event**

The mid-Holocene was characterized by overall less variability of the AMOC than compared to the early Holocene with minor changes of ± 2.5 ( ± 1.1) Sv, mostly within the uncertainty (Fig. 4). Following the series of multiple meltwater pulses that occurred in the Early Holocene around 8.2 ka BP, the AMOC rapidly recovered, reaching 18 Sv ( ± 0.8) between 8 and 7 ka BP, before gradually declining to approximately 16 Sv ( ± 0.8). Thereafter, the AMOC slightly increased toward PI levels, exhibiting a period of stability[54]. However, our by far best resolved record from ODP 1063 shows a slight but significant increase in $^{231}$Pa/$^{230}$Th between 4.5 and 3.8 ka BP (reaching a Z-score of 2.1, indicating that this peak is 2.1 SD above its mean), clearly standing out from its otherwise relatively stable $^{231}$Pa/$^{230}$Th (Fig. 2c), suggesting a potential ~15% AMOC weakening (Fig. 4b).

Notably, in the previously published dataset from ODP 1063 there was only a single data point deviating from the long-term Holocene mean towards higher $^{231}$Pa/$^{230}$Th[10]. Here, we first refined the age model of ODP 1063 by adding five new $^{14}$C tie points between 12.4 and 3.7 ka BP (Supplementary Table 2), so that the peak in $^{231}$Pa/$^{230}$Th is now dated to ~4.0 ka BP, while it was previously located at ~3.8 ka BP[10]. Second, further data points around this time period have been added to this record (Fig. 2c, light brown), supporting a $^{231}$Pa/$^{230}$Th increase.

The timing of this peak coincides with the 4.2 ka BP event, a well-documented severe climatic event defining the beginning of the late Holocene[55]. There is ample evidence for wet or dry climatic conditions in many regions all over the globe during this event. However, there is to date little understanding about the mechanism responsible for this event as well as about its global spread[56]. While a leading role of the North Atlantic and the AMOC seems possible for such a (nearly) globally recorded event with changes in the wind-fields, moisture transport, and temperature variations[57], there is no clear evidence for an oceanic origin of this event[58-60]. In fact, there is little supporting evidence for cool and/or ice-bearing surface waters penetrating well into the core of the North Atlantic Current at that time, as proposed on

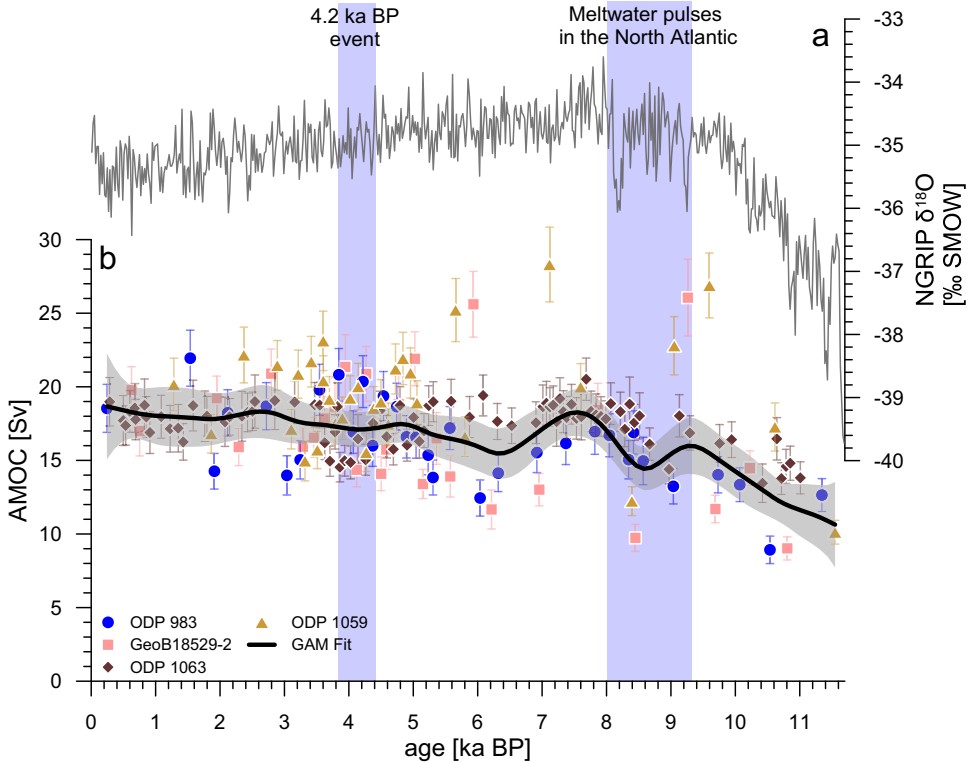

**Fig. 4 | Mean Holocene Atlantic Meridional Overturning Circulation (AMOC) strength (black line) with its 95% confidence interval (grey envelope). a** NGRIP (North Greenland Ice Core Project) $\delta^{18}O^{102}$. **b** GAM (Generalized Additive Model) fit of the Holocene AMOC strength. Data were calculated using the Bern3D quadratic relationships between the measured $^{231}Pa/^{230}Th$ (coloured data points, including their respective standard errors) and AMOC flux for each core location. The timing of multiple meltwater pulses in the North Atlantic around 8.2 ka BP[49] and the 4.2 ka BP event were highlighted with blue vertical bars.

the basis of ice-rafted debris depositions[37]. The lack of evidence for an oceanic signal around 4.2 ka BP might suggest that the AMOC did not play a crucial role in driving the globally observed terrestrial and socio-cultural impacts. Hence, mechanisms like solar variability, volcanic eruptions or short-term variability of the monsoon are potentially better candidates for causing changes in precipitation patterns, independent from AMOC variations[61]. Despite this, based on Earth system modelling, a slowdown of the AMOC has been recently invoked as the potential trigger for the physical processes responsible for the cold spell and mega-droughts during the 4.2 ka event[62]. Therefore, this peak at ODP 1063 could point towards a potentially overlooked connection (e.g., a non-deglacial related freshwater pulse) between ocean circulation and atmospheric perturbations[63,64]. However, the absolute $^{231}Pa/^{230}Th$ ratios during this event are comparable to those of the early Holocene after the YD at ODP 1063. Thus, these results would imply a similarly strong change in AMOC strength, compared to the YD[65], which would lead to a widespread and significant climatic perturbation. Yet, there is little to no evidence for such an AMOC reduction around this time from any other record or climate archive[59]. Further, a corresponding increase in $^{231}Pa/^{230}Th$ is missing in the other cores investigated in this study (Fig. 2), with some of them exhibiting the required temporal resolution and higher sedimentation rates than ODP 1063 during this time period. Between 5 and 4 ka BP $^{231}Pa/^{230}Th$ values as well as its variability are not distinguishable from the rest of the other Holocene records.

It therefore appears that an influencing factor other than the AMOC may have played a dominant role at ODP 1063 around 4.2 ka BP, possibly of a more local nature. This suggestion is supported by the fact that elevated $^{231}Pa/^{230}Th$ values are associated with higher $^{232}Th$-flux (Fig. 5), indicating an increased input of lithogenic material to the Bermuda Rise. Higher fluxes of lithogenic ($^{232}Th$ bearing) material imply airborne dust input and/or intensified sedimentation of

resuspended material out of the benthic nepheloid layers (BNL) present at this location[66,67]. Benthic storms, capable of producing pronounced nepheloid layers, are episodes of strong bottom currents and sediment resuspension that occur at abyssal depths often observed in regions with strong, eddying surface currents, such as the western North Atlantic[68]. Specifically, around the Gulf Stream system, high surface eddy kinetic energy can propagate downward, reaching the seafloor in the form of (anti)cyclones or topographic waves that are sufficiently strong to erode seafloor sediments and create benthic storms[69]. Thus, we suggest that the elevated $^{231}Pa/^{230}Th$ values around 4.2 ka BP at core ODP 1063 are the result of increased bottom scavenging of $^{231}Pa$ by intensively stirred nepheloid layers rather than an AMOC weakening signal. We suggest that an AMOC slowdown is unlikely to have triggered the 4.2 ka event in support of the perception that the 4.2 ka event is not a globally significant climate excursion[56]. Therefore, other more regional triggers like low-latitude forcing (e.g., tropical sea surface temperature anomalies) are more plausible[70,71]. Consequently, the reconstructed atmospheric disturbances during this event, which led to droughts and cold conditions[72], may have also induced eddy kinetic energy from the ocean surface to its depths, producing particularly strong BNL at ODP 1063.

**Late Holocene and future perspectives**

The AMOC remained largely constant from 6.5 ka BP with a slight increase over time until the PI. However, our data, constrained by the limitations inherent to both sedimentary archives and the $^{231}Pa/^{230}Th$ proxy, would not allow for the detection of AMOC changes as observed today during anthropogenic climate change, simply due to the available observation period from sedimentary $^{231}Pa/^{230}Th$. Nevertheless, the contrast between projected future AMOC scenarios and the relatively constant AMOC over the last 6,500 years is evident:

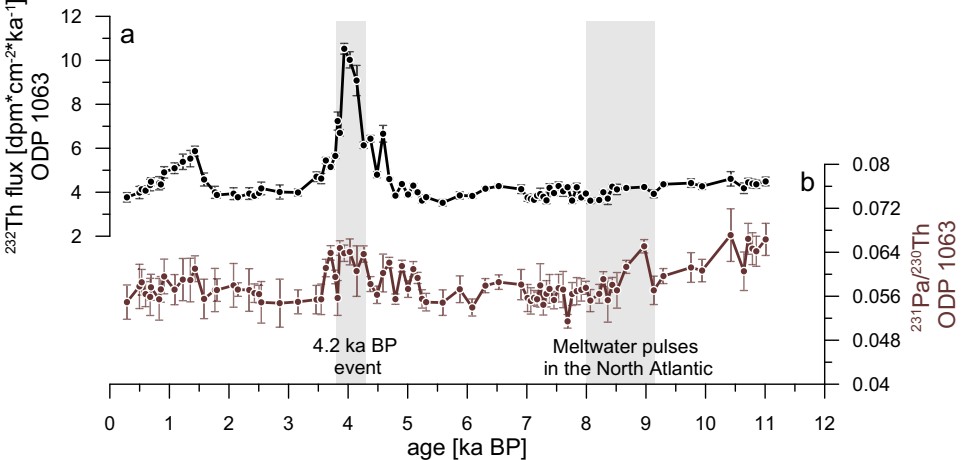

**Fig. 5 | Holocene lithogenic flux at Bermuda Rise.** $^{230}$Th-normalised $^{232}$Th-flux(**a**) (black line), compared with the $^{231}$Pa/$^{230}$Th of ODP Site 1063 (**b**) (brown line)[10]. The error bars indicate the standard errors associated with each data point. Grey bars highlight the timing of multiple meltwater pulses in the North Atlantic around 8.2 ka BP and the 4.2 ka BP event.

Under the optimistic CMIP6 (Coupled Model Intercomparison Project) SSP1-2.6 (Shared Socioeconomic Pathways) scenario, representing a global warming by 2100 CE of around 1.5 °C, the AMOC is projected to decrease by about 5 Sv[73]. AMOC changes in the Holocene, similar to those simulated for the optimistic scenarios (i.e., SSP1-2.6 and 2–4.5), cannot be ruled out on the basis of our data alone, due to the limitations of our approach. On the other hand, the high-emission SSP5-8.5 and SSP3-7 scenarios projects a global warming of 3 to 5 °C by 2100 CE, and an AMOC decrease of around 8 Sv[73]. In this case we would expect to be able to detect such hypothetical AMOC signals (CMIP6 SSP3-7 and 5-8.5) until 2100 to 2200 CE. Thus, based on the results of this study we can rule out that similar changes in AMOC strength, as pessimistically projected for the future, have happened within at least the last 6500 years. This implies that, if these projections for the future AMOC are correct, it would mark an unprecedented change of the long-term AMOC, regime after approximately 6500 years of relative stability.

In summary, this study presents a high-temporal resolution quantitative AMOC reconstruction for the Holocene by converting multiple sedimentary $^{231}$Pa/$^{230}$Th records into volumetric flow rate using the Bern3D model. Through idealized simulations applying varying freshwater perturbations, the sensitivity of $^{231}$Pa/$^{230}$Th to AMOC changes was assessed. Based on the response of $^{231}$Pa/$^{230}$Th at individual core locations to a wide range of AMOC strengths, a new composite record converting $^{231}$Pa/$^{230}$Th into AMOC strength for the Holocene has been generated. Our composite record reflects the mean large-scale AMOC strength in the western North Atlantic during the Holocene. It shows the weakest AMOC at the onset of the early Holocene, that is about 8 (±2) Sv weaker than the PI AMOC. This slowdown is likely a result of the aftermath of the YD. A second period of weaker AMOC, about 3 (±1.3) Sv less relative to PI levels, occurred from 9.2 to 8 ka BP. This slowdown coincides with several pronounced meltwater pulses in the North Atlantic, like the one associated with the 9.2 and 8.2 ka event, presumably originating from meltwater of the LIS. However, due to bioturbation blurring the $^{231}$Pa/$^{230}$Th signal, the 8.2 ka event cannot be specifically identified from a general AMOC weakening between 9 and 8 ka BP. Our results indicate that the highest Holocene fluctuations in AMOC strength occurred within the early Holocene (11.7–8.2 ka BP). These fluctuations diminished throughout the mid-Holocene (11.7–4.2 ka BP), stabilizing after 6.5 ka BP, beyond which the AMOC exhibited a high degree of long-term stability. Nevertheless, hypothetical AMOC changes on centennial time scales cannot be definitively identified from this record, due to the intrinsic limitations in temporal resolution of the $^{231}$Pa/$^{230}$Th proxy. Consequently, hypothetical short-term events (shorter than ~180 years) of significant AMOC anomalies (weaker than 8.5 Sv) cannot be ruled out for the Holocene. However, given the stable AMOC conditions inferred from our analysis for the past 6,500 years, projections on the magnitude of modern AMOC slowdown until the end of the century may indeed reflect an unprecedented change of the long-term Holocene AMOC regime.

## Methods

### Marine sediments and core locations

The northernmost site of the studied transect is ODP 983 (60.40°N, 23.64°W; 1984 m water depth), situated within the rapidly accumulating Gardar Drift (GD) south of the Iceland-Faroe-Ridge (IFR). The sediment accumulation at this location is dependent on the dense water masses flowing over the IFR, a process that is in turn influenced by the general climate of this region[74]. As a result, ODP 983 has been subject to detailed investigation in the past[75,76].

Further south, on the edge of the Grand Banks Margin (GBM) within the Newfoundland Basin, core GeoB18529-2 (41.9°N, 47.6°W; 3989 m water depth) was recovered, approximately 3500 km off the coast of Newfoundland. This Newfoundland Basin is neighboured by the Labrador Basin to the north, the North Atlantic Ridge to the east and the North American Basin to the Southwest. Consequently, the site is not directly influenced by the continental margin, thereby offering insights into open-ocean environmental conditions.

Even further to the south, within the confluence region of North Atlantic Deep Water (NADW) and Antarctic Bottom Water (AABW), lies the Bermuda Rise (BR), from where sediment cores ODP 1063 (33.7°N, 57.6°W; 4584 m water depth[10]) and OCE326-GGC5 (33.7°N, 57.6°W; 4550 m water depth [15]) were collected. The BR is characterised by a high sensitivity to past variations in ocean currents[77,78]. Consequently, both BR sediment cores have been extensively investigated[10,13,17,24,79]. Additionally to the already existing ODP 1063 $^{231}$Pa/$^{230}$Th dataset[10], another 17 data points of this site have been generated within this study, enhancing the temporal resolution within particularly interesting time intervals. Both sites of the BR are located in close proximity to each other and are further considered as a single combined core.

The southernmost cores investigated in this work, KN140-2-51GGC (32.8°N, 76.3°W; 1790 m water depth) and ODP 1059 (31.7°N, 75.4°W; 2985 m water depth), are situated at the Blake Bahamas Outer Ridge (BBOR). The BBOR is a well-characterised sediment drift deposit, ideally

suited for high-resolution paleoceanographic reconstructions[80]. This study uses the results of KN140-2-51GGC[9] and ODP 1059[81]. The previously available $^{231}$Pa/$^{230}$Th record from ODP 1059 was extended by 25 data points covering the Holocene.

Overall, the $^{231}$Pa/$^{230}$Th records of ODP 983, GeoB18529-2 and ODP 1059 are sensitive to past AMOC variations and are located at different water depths compared to the two previously published high-resolution records from the Bermuda Rise (BR) (ODP 1063 and OCE326-GGC5[24,82] and from the Blake Bahamas Outer Ridge (BBOR) (KN140-2-51GGC[9]) (Fig. 2).

In combination, these five locations collectively form a transect spanning vast swaths of the western North Atlantic, encompassing water depths ranging from 1790 to 4584 m (Fig. 1). Sedimentation rates of the newly investigated cores range from 9 to 23 cm/ka (Supplementary Table 1).

## Age models

We have adopted the chronologies of ODP 983, KN140-2-51GGC, and OCE326-GGC5 and employed the available age models[9,24,83,84]. The existing radiocarbon-based Holocene age models for ODP 1063[10] and ODP 1059[82] were refined with additional radiocarbon ages and recalibrated using the latest radiocarbon calibration curve (Supplementary Table 2). For GeoB18529-2, which lacked a published Holocene age model, we developed a new chronology based on seven $^{14}$C ages (Supplementary Fig. 5).

For ODP 1063, five radiocarbon ages were added to the 13 previously published ones[10], all derived from mixed planktic foraminifera[85]. Similarly, eight radiocarbon ages were generated for ODP 1059 using mixed planktic foraminifera[85]. For GeoB18529-2, we obtained seven radiocarbon ages from *Globigerina bulloides* and *Neogloboquadrina pachyderma* (sinistral)[85]. In total 20 new $^{14}$C ages were obtained via accelerator mass spectrometry (AMS) at the University of Bern, ETH Zürich, and by Beta Analytic Inc. An overview of these $^{14}$C ages, along with their uncertainties (reported as 1 sigma standard deviations), is provided in Supplementary Table 2.

All radiocarbon ages were calibrated with Marine20[86], using the rbacon tool[87]. To account for the local marine reservoir effect, we applied a weighted mean of ten location-specific reservoir age offsets (ΔR) (Supplementary Table 3). Based on these calibrated and reservoir-corrected radiocarbon ages (Supplementary Table 2), and by using rbacon v3.3.1 (settings provided in Supplementary Table 3), we recalculated the already published $^{14}$C-constrained age models for ODP 1063 (Supplementary Fig. 6 & 7) and ODP 1059 (Supplementary Fig. 9 & 10) following[10] and[82], respectively. For core GeoB18529-2, we generated a new age model (Supplementary Fig. 5). Further details on the age models are provided in the Supplement.

## Uranium, thorium, and protactinium isotope analyses

Sediment samples from the core sites were analysed for the radioisotopes $^{230}$Th, $^{231}$Pa, $^{232}$Th, $^{234}$U, and $^{238}$U. Per sample approximately 0.1 g of sediment was weighed and then spiked with $^{233}$Pa, $^{229}$Th and $^{236}$U prior to chemical treatment, followed by total digestion in a mixture of concentrated HCl, HNO$_3$ and HF. Purification and separation of Pa, Th and U followed the standard protocols[88]. The short-lived $^{233}$Pa spike ($t_{1/2} = 27$ d) was milked from a $^{237}$Np solution using silica resin and a mixture of HNO$_3$ and diluted HF[89]. The $^{233}$Pa spike was calibrated against an internal pitchblende standard[90] and the reference material IAEA-385[88] as well as the concentrations of $^{230}$Th, $^{232}$Th, $^{234}$U, and $^{238}$U in the samples. Isotope measurements were performed on a Neptune Plus MC-ICP-MS in the Geozentrum Nordbayern at the Friedrich–Alexander University in Erlangen equipped with a retarding potential quadrupole filter, on an ELEMENT ICP-MS at the AWI Bremerhaven and on an iCAP TQe ICP-MS at the Institute of Earth Sciences of Heidelberg University. Full process blank contributions to the final concentrations of $^{230}$Th, $^{232}$Th and $^{238}$U were generally lower than 1%

and below 2% for $^{231}$Pa. The excess fractions (that is the $^{231}$Pa and $^{230}$Th produced in the overlying water column from the decay of dissolved uranium) were calculated from the total concentrations corrected for detrital and authigenic input and decay corrected since the time of deposition[91]. The detrital correction ($^{238}$U/$^{232}$Th) for each core is based on the overall minima of bulk $^{238}$U/$^{232}$Th for each core. Due to the relatively young age of the examined samples both, the authigenic and the lithogenic contributions are of relatively low importance for the total concentrations compared to the excess fraction.

## Biogenic opal concentrations

Very high $^{231}$Pa/$^{230}$Th ratios, as observed in the Southern Ocean, are the result of two different processes. First the import of excess $^{231}$Pa from the North-Atlantic and second the high proportion of biogenic opal among the particles with its high affinity to $^{231}$Pa[92]. This particular influence of opal on $^{231}$Pa/$^{230}$Th requires variations in a $^{231}$Pa/$^{230}$Th down-core profile to be checked for potential influence of opal before they can be interpreted as direct result of AMOC variability[45,93]. Accordingly, we have measured the biogenic opal (bOpal) content of each sample to assess potential impacts of the particles flux on the $^{231}$Pa/$^{230}$Th. These measurements were performed following the automatic procedure for analysis of dissolved silica applying molybdate-blue spectrophotometry[94] at the Institute of Earth Sciences of Heidelberg University. The dried and milled sample is first dissolved using Na$_2$CO$_3$. Then, ammoniumheptamolybdate is added to the sample, resulting in the formation of yellow molybdosilicic acid. Any excess ammoniumheptamolybdate is bound by the addition of oxalic acid. Lastly, ascorbic acid is introduced, causing the yellow molybdosilicic acid to transform into silicomolybdenum blue, which is subsequently measured using photometry. In order to convert the measured absorbance into the bOpal concentration, a calibration was performed using Certipur® Silicium Standard Solution. Besides this calibration, an internal reference material with about 3% bOpal (LOW-PAL) content was repeatedly measured for quality control and for defining the reproducibility of the method.

## Bern3D model

To estimate the AMOC variations associated with changes in reconstructed $^{231}$Pa/$^{230}$Th we here employ the $^{231}$Pa/$^{230}$Th-enabled Bern3D Earth system model of intermediate complexity v3.0[95]. The Bern3D model comprises a dynamic geostrophic-frictional balance ocean module, a single-layer energy-moisture balance atmosphere, and a biogeochemistry module that simulates the carbon cycle as well as geochemical proxies. The horizontal resolution of the ocean and atmosphere components is 68×46 grid cells, and 40 logarithmically scaled depth layers in the ocean. $^{231}$Pa and $^{230}$Th are implemented in the model as two tracers for each isotope, representing the dissolved and particulate fractions[29]. Further modifications to this implementation are reported in[47]. In brief, the explicit parameterization of boundary scavenging was removed as it is implicitly included in the reversible scavenging. Bottom scavenging by benthic nepheloid layers is explicitly accounted for by prescribing modern nepheloid layer data (i.e., a static field). Importantly, scavenging is now particle concentration-dependent, as observed in the modern ocean. Tracer concentrations are calibrated against modern seawater data of the GEOTRACES program[96]. For this, scavenging factors and desorption parameters were sampled with a Latin hypercube approach to capture the parameter space as efficiently as possible and were then calibrated against modern dissolved concentrations of Pa and Th. More details on the tracer tuning can be found in[47]. A detailed description of the physical state of the model can be found in[95] including a description of model biases.

To estimate the detectability of AMOC changes in simulated $^{231}$Pa/$^{230}$Th at the specific core sites for which $^{231}$Pa/$^{230}$Th was reconstructed in this study, we performed a range of idealized simulations (Fig. 3). First, the model was spun up for 45000 years under PI conditions until an equilibrium state was reached. Greenhouse gas

Article

concentrations are set to $CO_2 = 278.05$ ppm, $CH_4 = 721.89$ ppm, $N_2O = 272.96$ ppm and $^{231}Pa/^{230}Th$ tracers are only included after 35150 years of spin up. These simulations were performed under pre-industrial boundary conditions. Freshwater was applied to the North Atlantic between 45°N and 70°N over time periods of 100, 120, 140, 160, 180, 200, and 300 years linearly increasing for the first half of the forcing period and then linearly scaled back (triangular shape). The amount of freshwater was chosen such that the maximum AMOC weakening is −4.9, −6.8, −8.5 Sv which corresponds to maximum freshwater forcings of 0.04, 0.082, and 0.15 Sv. The AMOC evolution of all simulations is depicted in Supplementary Fig. 3 including the $^{231}Pa/^{230}Th$ responses at the specific core sites (21 different scenarios). Site-specific factors like sedimentation rate (Supplementary Table 1), water depth (Supplementary Table 1), sampling interval (1 cm) and bioturbation (5 cm depth) were considered of each core location for these simulations.

Next, we assessed the sensitivity of $^{231}Pa/^{230}Th$ to AMOC changes, under PI boundary conditions, with a suite of 20 idealized simulations (Supplementary Fig. 12). In each simulation, the AMOC is forced to a rapid increase in strength and gradual slowdown of different lengths (500–2000 years) and amplitudes (4–23 Sv) by applying varying freshwater perturbations in a latitudinal band between 45°N and 70°N in the North Atlantic (Supplementary Fig. 13). For verification, the stream function of the AMOC for the PI state, as well as for the strongest and weakest states, has been included in Supplementary Fig. 14. Thereby, we obtain a wide range of AMOC strengths and corresponding $^{231}Pa/^{230}Th$ values.

To estimate the core-specific sensitivity of $^{231}Pa/^{230}Th$ to AMOC variations, we identify for each core location the corresponding grid cell in Bern3D and extract the temporal $^{231}Pa/^{230}Th$ evolution as an average over all neighboring grid cells to smooth out local model biases. The extracted $^{231}Pa/^{230}Th$ evolutions are low-pass filtered and subsampled to create bioturbated pseudo-proxy records with a bioturbation depth of 5 cm and a sampling interval of 1 cm based on the core-specific sedimentation rates (Supplementary Table 1). The generated pseudo-proxy $^{231}Pa/^{230}Th$ records are set relative to the ratios representing the modern (PI) $^{231}Pa/^{230}Th$ value (resulting in $\Delta^{231}Pa/^{230}Th$) and then plotted against the respective AMOC evolutions (Supplementary Fig. 12). The core-specific relationship between the modelled AMOC strength and $^{231}Pa/^{230}Th$ (red dashed line) is assessed with a least square's polynomial fit of degree 2. The resulting relationships are given in Supplementary Table 4.

Using these model-based relationships between modeled AMOC strength and $^{231}Pa/^{230}Th$ for every individual core site, we converted the sedimentary $^{231}Pa/^{230}Th$ from all cores into one timeseries of past AMOC strength (in Sv). In this conversion, the error propagation was addressed by combining the analytical uncertainty of the sedimentary $^{231}Pa/^{230}Th$ ratios (Supplementary Data Sheet) with the uncertainty of the respective simulated relationship between $^{231}Pa/^{230}Th$ and AMOC strength (Supplementary Table 4). A GAM fit[97] was applied to the Sverdrup data to identify shared trends across the different core sites. To ensure that this GAM fit reflects the uncertainty of the data, an error weighting ($1/error^2$) was incorporated. Additionally, the 95% confidence interval was computed and visualized as a grey error envelope (Fig. 4).

## Data availability
The data generated in this study have been deposited in the PANGAEA repository for Earth & Environmental Science database under the accession codes https://doi.org/10.1594/PANGAEA.980737 (radiogenic isotopes and bOpal data)[98], https://doi.org/10.1594/PANGAEA.980738 (AMOC strength estimates)[99] and https://doi.org/10.1594/PANGAEA.980739 ($^{14}C$ data)[85].

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

## Acknowledgements

This research used sediment samples provided by the Ocean Discovery Program (ODP), which is supported by NSF and participating countries under management of Joint Oceanographic Institutions (JOI) Inc. We thank Walter Geibert and Ingrid Stimac for analytical support and provision of infrastructure, Sharon Hoffmann for providing data and Patrick Blaser for discussions. This study was funded by the Deutsche Forschungsgemeinschaft (DFG, German Research Foundation – 277128673). FP was financially supported by the European Union's Horizon 2020 research and innovation program under Grant Agreement No 101023443 (project CliMoTran). CMC acknowledges the financial support from FAPESP (grant 2018/15123–4), CNPq (grants 406898/2022-7 and 305285/2025-4), and CAPES-COFECUB (grants 8881.712022/2022-1 and 49558SM).

## Author contributions

L.G., J.L., F.S., O.V., M.E., S.T., M.R., L.M., C.C., and S.S. performed the analyses. L.G., J.L., F.S., P.T., and F.P. interpreted the data. P.T and F.P. performed model experiments. L.G., J.L., F.P., O.F., and C.C. wrote the paper with input from F.S., O.V., P.T., M.E., S.T., L.M., M.R., S.S., and J.L. designed the study.

## Funding

## Competing interests

The authors declare no competing interests.
