## [Transparent Peer Review file · Nature Communications]

Low variability of the Atlantic Meridional Overturning Circulation throughout the Holocene

Corresponding Author: Mr Lukas Gerber

Version 0:

Reviewer comments:

Reviewer #1

(Remarks to the Author)

Reviewer #2

(Remarks to the Author)

Gerber et al. investigates variability in Atlantic Meridional Overturning Circulation (AMOC) during the Holocene (last 11.7 ka) from two entirely new sedimentary $^{231}\text{Pa}/^{230}\text{Th}$ records from the North Atlantic, four existing mid-latitude records, and using Bern3D, an Earth System Model of intermediate complexity. This is an important topic, as the AMOC has been identified as a core climate tipping element in the Earth system (e.g. Armstrong McKay et al. 2022, Science); IPCC AR6 has stated it is very likely the AMOC will decline over the 21st century, but there is low confidence in the quantitative projections due to poor process understanding (Fox-Kemper et al., 2021).

The new proxy reconstructions presented in this study address a highly uncertain, but important topic with respect to the global climate. This work could be a useful contribution to Nature Communications. The main findings are that fluctuations in AMOC strength were highest in the early Holocene as the ice sheets deglaciated, with a pronounced slowdown during the 8.2 ka BP meltwater pulse event, and centennial-scale AMOC variability remained low since 6.5 ka BP, with no notable change at 4.2 ka BP.

However, I note that these results are model-dependent because the interpretation of the records, and in particular the differences between the various records, is based on simulations with one intermediate complexity model. I have some concerns with respect to the modelling aspect of this work that I hope the authors can address. This review is focused on the Earth system modelling as the proxy reconstruction is outside of my particular expertise.

General comments

The experimental design for the idealised Bern3D simulations make sense, but the methods description is vague and lacking detail. As the methods are currently written, I am convinced I would not be able to set up and run these same experiments without more information, so the authors should consider the reproducibility of these simulations. As some examples for more detail: What was the spin-up procedure? How was the model validated? What volumes of freshwater input were applied in each of the idealised simulations (AMOC weakening of 4.8, 6.8, and 8.5 Sv), and what were these amounts based on?

The authors should do more to demonstrate that their proxy interpretation is not dependent on this particular intermediate complexity model, or at least better contextualise their results noting the diversity of AMOC representation in models of various levels of complexity. Highlighting the modelling challenge, Weijer et al. (2020) show the wide range of AMOC strength and response to SSP scenarios in CMIP6 models; He and Clark (2022) demonstrate that some AOGCMs may be too sensitive to freshwater input; Obase et al. (preprint) show substantial differences in AMOC strength among PMIP4 models over the last deglaciation. In this study, the idealised simulations are used to investigate the detectability of AMOC

perturbations at the individual sites. The main question I'd like the authors to consider here is: How would the proxy interpretation change if using a different ESM?

Specific comments

Line 1: Should the title be "multi-centennial"? Just noting the temporal resolution of the records is 0.5 ka.

Line 73-105: I find this paragraph difficult to follow because it covers so many topics: how the $^{231}\text{Pa}/^{230}\text{Th}$ has been used previously; 8.2 and 4.2 ka BP events; sedimentological conditions at particular sites. I recommend revising for brevity and clarity.

Line 90: Northern Hemisphere

Line 92: Noting two ".."

Figure 1: Specify which records are "new"

Figure 2: Triangle colours need explaining. Are there no age model tie points for ODP 983?

Line 185: Double check grammar of this sentence.

Line 517: Do you consider the impact of freshwater flux in the Southern Ocean (e.g. Bakker et al. 2017)? I recommend performing additional sensitivity experiments with Bern3D to investigate how site detectability changes with a Southern Hemisphere-sourced AMOC perturbation.

Supplementary Information: I recommend adding a figure showing the streamfunction fields for the Preindustrial control and the AMOC perturbation experiments (e.g. see Boulton et al., 2014, Nature Comms., Fig 1).

References

Armstrong McKay, D.I. et al. Exceeding 1.5°C global warming could trigger multiple climate tipping points. *Science* 377, eabn7950(2022). DOI:10.1126/science.abn7950

Fox-Kemper, B., et al. Ocean, Cryosphere and Sea Level Change. In *Climate Change 2021: The Physical Science Basis. Contribution of Working Group I to the Sixth Assessment Report of the Intergovernmental Panel on Climate Change*. Cambridge University Press, Cambridge, United Kingdom and New York, NY, USA, pp. 1211–1362, (2021). doi: 10.1017/9781009157896.011.

Weijer, W., et al. CMIP6 models predict significant 21st century decline of the Atlantic meridional overturning circulation. *Geophysical Research Letters* 47.12 (2020): e2019GL086075.

He, F., Clark, P.U. Freshwater forcing of the Atlantic Meridional Overturning Circulation revisited. *Nat. Clim. Chang.* 12, 449–454 (2022). <https://doi.org/10.1038/s41558-022-01328-2>

Obase, T., Menviel, L., Abe-Ouchi, A., et al. Multi-model assessment of the deglacial climatic evolution at high southern latitudes, *Clim. Past Discuss.* [preprint], <https://doi.org/10.5194/cp-2023-86>, in review, 2023.

Bakker, P., Clark, P., Golledge, N. et al. Centennial-scale Holocene climate variations amplified by Antarctic Ice Sheet discharge. *Nature* 541, 72–76 (2017). <https://doi.org/10.1038/nature20582>

Boulton, C., Allison, L. & Lenton, T. Early warning signals of Atlantic Meridional Overturning Circulation collapse in a fully coupled climate model. *Nat Commun* 5, 5752 (2014). <https://doi.org/10.1038/ncomms6752>

Reviewer #3

(Remarks to the Author)

Gerber and coauthors analyzed multiple high-resolution $^{231}\text{Pa}/^{230}\text{Th}$ sedimentary records of the North Atlantic. The authors also used climate model simulations to quantify past AMOC strength by addressing the different depths and locations of sedimentary records. They estimated the strength of AMOC during the Holocene, particularly for the minimum value of the Holocene and trends.

The materials and methods seem good, but I still don't understand some points related to climate model simulations, which is important in estimating quantitative AMOC strength changes. I hope the revised manuscript clarifies the following points.

L195-202: These lines correspond to explanations of Figure 3 and Supplementary Figure 3, which makes sense. However, the method section says the 20 idealized simulations with different lengths and amplitudes by applying varying freshwater perturbations for the Bern3D model (L514-519). Figure 3 shows that the minimum values of the AMOC in seven rows are identical (-4.9, -6.8, -8.5Sv), so I assume the 20 simulations are scaled to create 21 scenarios but not sure. Please explain how to get 21 AMOC time series in Figs.3 and S3 from the original 20 Bern3D model simulations.

Figure 6 needs the derivation of individual blue dots and methods to get dashed red lines. And I couldn't find the reference to Figure 6 in the manuscript.

Version 1:

Reviewer comments:

Reviewer #2

(Remarks to the Author)

This is the second round review of Gerber et al. "Low centennial-scale variability of the Atlantic Meridional Overturning Circulation throughout the Holocene". To reiterate my previous review, this is an important topic given that the AMOC has been identified as a core climate tipping element, but with low confidence in quantitative projections. This record of Holocene AMOC variability is therefore an important contribution.

In this revised manuscript, I appreciate the authors' efforts at addressing my previous concerns. The title of the manuscript is now representative of the findings, the introduction is much improved, and I found the revised methods section for Bern3D model very clear. The authors adequately addressed my concern regarding freshwater forcing, clarifying that these experiments are idealised to show the effect of the AMOC perturbation, but still noting the values, which allows for comparison to other models.

Reviewer #3

(Remarks to the Author)

Reviewer #4

(Remarks to the Author)

Review of "Low variability of the Atlantic Meridional Overturning Circulation throughout the Holocene" Gerber et al Nature Communications

This manuscript presents Holocene reconstructions of AMOC generated with Pa/Th. Pa/Th is one of the less intuitive proxies, so using an intermediate complexity earth system model really helps to interpret the results.

Despite the relative insensitivity of the proxy system to moderate changes in AMOC, I think the conclusions that anthropogenic driven changes will be larger than anything in the Holocene is reasonable.

The authors seem to have addressed concerns raised by the reviews in round one.

Now that the ms focuses on the millennial scale trends (the previous claim of centennial scale was unsupportable), the demands on the age-depth models are reduced. This is fortunate. The age-depth models are sufficient for the current work, but not for claims of centennial scale.

The ms reports that it improves the age-depth model for ODP 1063 by adding 5 extra dates but does not show the resulting model so the reader cannot evaluate how reliable it is. I cannot tell what type of model was used (and the the original paper for this core does not help). I don't understand why intcal20 was used - is this a typo - marine20 would seem to be a more appropriate calibration curve. I presume that the dates are being archived on pangaea, but this isn't explicitly stated (would be helpful to see supplementary data sheet which apparently lists the data).

Extra dates are also included for ODP 1059. Again insufficient information is given to evaluate the chronology.

A new age-depth model is made for GeoB18529-2. No information is given for the bacon settings. I presume the default priors and other settings are used. This is probably reasonable for this high-sedimentation site, and not reporting this information is not unusual. Again, I don't understand why intcal20 is being used on marine dates. The age-depth model for this core is shown. It is not the happiest chronology I have seen: two of the seven dates are completely outside model's 95% uncertainty. It is possible that the sedimentation rate is more varied than the current model suggests. Only more dates will improve the model.

Fortunately, because the reconstructed AMOC is not very dynamic, the demands on the chronology are not too great.

Other issues

line 194 "partly very different" rephrase - perhaps "occasionally very different"

Fig 4 (with implications for the other figures). I find the red and orange quite difficult to distinguish. Please try to increase the contrast between colours and consider using shape to help distinguish the points.

line 347 "6.5 ka BP on until the PI" delete "on"

line 468 "for 231Pa below 2 %" -> and below 2% for 231Pa

Fig S3. The time axis on this plot runs from left to right, whereas on the other plots, time runs right to left.

The scale for the grey line in the upper panels is not explicitly given.

I don't fully understand the detrital correction. Is it possible that the increased influx of detritus at 4.2 ka in ODP 1063 is biasing the Pa/Th record?

Version 2:

Reviewer comments:

Reviewer #4

(Remarks to the Author)

Review of "Low variability of the Atlantic Meridional Overturning Circulation throughout the Holocene" Gerber et al

The revisions since the previous version have greatly improved the presentation of the chronology. However, I am still not convinced by the approach taken to calibrate the radiocarbon dates, but concede that the differences are probably minor.

The standard way to calibrate marine radiocarbon dates is to use the marine calibration curve, derived from the atmospheric calibration curve using a highly simplified carbon model, and a delta R value to account for the deviation between the modelled reservoir effect and the local conditions. The challenge is that delta R varies in time and space and for many areas it is poorly constrained.

An alternative would be to use a more complex ocean model forced by palaeoclimate to generate a bespoke calibration curve for each core location. The challenge is that the model is much slower and so cannot be run many times to explore its uncertainty.

With both these approaches, the resulting calibration curve is smoothed compared with the atmospheric curve.

I understand that for this manuscript, the reservoir age estimated with a relatively complex earth system model is being used with the atmospheric calibration curve. I don't think this is appropriate as there is no smoothing of the calibration curve that is inherent in the real ocean and in the carbon models. This can possibly be seen in some of the calibrated dates that have complex PDFs, which you don't normally see for marine dates given the smoothness of the marine calibration curve.

Assuming a bespoke calibration curve with uncertainties cannot be generated, a more appropriate method would be to use the delta-R from the complex model with marine20. I'm not quite sure how the delta-R would be defined from the model (perhaps against the modelled global mean).

As the authors show in the response to reviewers, the choice of methods does not have a large effect, but I think it is important to use the most reliable methods.

Minor points

The abbreviation mbsf does not seem to be defined.

The SI (and the response to reviewers) describes a 97.5% confidence interval. This is an unusual interval to choose, and differs from the default 95% interval (at least in rBacon). The 95% interval spans from 2.5% to 97.5%. Please check the interval has been reported correctly.

Table S2 "Age dated" and "Age dated sd" are strangely worded. Perhaps "¹⁴C age". The calibrated age range could be presented in one column rather than two.

Fig S10. Something is strange with the first radiocarbon data - the plotted PDF is a triangle that does not include the weighted mean and does not correspond to the uncertainty whiskers. Please check this figure has rendered correctly.

**Authors response to reviews on the manuscript entitled “Low variability of the Atlantic**
**Meridional Overturning Circulation throughout the Holocene” (NCOMMS-24-61663)**

We sincerely thank the Reviewers for their valuable and helpful inputs. We believe, that the revised version
of the manuscript improved substantially thanks to their feedbacks. Besides the specific points raised by
the Reviewers, we recognized two major topics that deserved improvement, namely:

the temporal resolution of our analysis, which we described previously as “centennial” (I), and the
description and explanation of our Bern3D model approaches (II). We will address these topics in detail
throughout our responses but wish to provide an overview of the revisions we have made upfront:

(I) We have revised our description of temporal resolution as we agree, that the term “centennial”
resolution is not accurate and potentially misleading. Therefore, we have decided to refrain from using
“centennial” resolution when discussing our GAM results.

(II) We have improved the description and explanation of our modeling approaches. We would like to
emphasize, that two distinct approaches were employed in this study:

First, we used Bern3D model simulations to assess the detectability of various AMOC slowdowns when
using $^{231}\text{Pa}/^{230}\text{Th}$ (Fig. 3). Separately from the first approach, we simulated core-specific sensitivities of
$^{231}\text{Pa}/^{230}\text{Th}$ to idealized AMOC slowdowns using a pseudo-proxy approach (Supplementary Fig. 6). Using
these AMOC-to- $^{231}\text{Pa}/^{230}\text{Th}$ relationships, we were able to convert our geochemical data into AMOC
strength (Sverdrup units) (Fig. 4). In this regard, we decided to move Fig. 6 of the previous version of the
manuscript to the Supplement (now Supplementary Fig. 6 in the revised manuscript) and expand our
explanation about this approach in the Methods section.

We have also decided to add Oliver Friedrich from the Institute of Earth Sciences at Heidelberg University
as a co-author, as he contributed to the interpretation and provided significant feedback on the
manuscript’s revisions and corrections.

In the following point-by-point response, the Reviewers’ comments are given in black, while our answers
are provided in blue. Stated line numbers refer to the revised versions of the manuscript and the
Supplement without “tracked changes”. Additionally, we are also providing a “tracked changes” version of
both documents, so that it is possible to follow any individual changes made by us in detail.

Citations from the revised manuscript are marked by “ ”, combined with its respective line reference.

We have subdivided the reviewer’s comments into individual points (numbering reviewer and enumerated
comments) in order to easier allow addressing and referring to related issues. As some comments from
the reviewers partly overlap, we respectfully refer to our responses to similar questions raised by other
reviewers wherever applicable.

We hope that our revisions and clarifications will fulfill the editors’ as well as the reviewers’ requirements
and expectations.

**Reviewer #1**

This paper is proposing a new reconstruction of the AMOC over the Holocene, based on 5 Pa/Th proxy
records in the Atlantic Ocean and scaled towards AMOC reconstruction using Bern3D model. The
reconstruction obtained is showing quite few variations over the Holocene. It is then highlighted that
given the density of proxy records and their time resolution, it is not possible to detect AMOC weakening
larger than about 5 Sv on a time frame shorter to about 200 years. This paper is based on a robust
approach and useful proxy records for reconstruction of the AMOC. In this respect this is a nice paper.

We sincerely thank Reviewer#1 for their review and constructive feedback on our work. We appreciate
the critical assessment of the description of our temporal resolution and also for bringing up the pseudo-
proxy approach.

For this point-by-pointy response, we chose to split up some comments to allow for a clearer and more
detailed response.

R1.1

However, the statements from the title and some others throughout the paper are just disagreeing with
the evidences provided in the paper. For instance, Fig. 3, which is the climax of the paper in my view, is
just demonstrating that the approach cannot allow to reconstruct centennial variability, except for time
scale larger than 200 years and AMOC changes of very substantial amplitude (about more than 30% of
the present-day amplitude!). Thus, while this reconstruction is of interest, it cannot, by construction,
supports the claims of the paper.

What this reconstruction can solve has more to do with millennial variability. To solve centennial
variability, you actually need to have proxy records with a time resolution about one order of magnitude
lower than the period you are analysing. This can be simply understood in the scheme provided in this
review (Fig. R1). To solve a 100-yr cycle, you need to be able to sample the high phase and low phase of
the cycle, which are separated by about 50 years. To have robust results, I think (and this can be proven
properly, see specific comments) you might need resolution of about 10 years, which is far from the
resolution of proxy records used. Due to this editorial flaw (why discussing centennial variability while
your records do not really allow to access it?), I cannot support the publication of this paper in its present
form.

We appreciate the detailed comments.

The mean temporal resolutions for the individual \$^{231}\text{Pa}/^{230}\text{Th}\$ records range from 2.4 (GeoB18529-2) to 7.4
(ODP 1063) samples per 1000 years. This information is now provided in Supplementary Table 1. We note
that our individual records have thus a millennial-scale temporal resolution.

However, the GAM fit consists of in total 158 \$^{231}\text{Pa}/^{230}\text{Th}\$ data points (not considering the data from core
KN140-2-51GGC data). By employing our model data approach, which effectively combines the individual
records (besides KN140-2-51GGC, Fig. 3) into one large scale AMOC estimate, we have an average sampling
resolution of \$\sim 13.5\$ samples per 1000 years. This has led us to discuss 'centennial variability', given that we
were able to resolve the 8.2 ka BP event under certain assumptions. Even though there are different uses
of the term 'centennial time resolution' out in the literature, we agree that this term, in particular in the
title, could be misleading.

We fully understand this criticism. Based on these suggestions of Reviewer#1 (R1.1), Reviewer#2 (R2.6)
and the data from Supplementary Table 1, we now refer to the mean temporal resolution of our
sedimentary \$^{231}\text{Pa}/^{230}\text{Th}\$ records as 'millennial', while referring to the results of the GAM fit as 'sub-
millennial' (lines 1, 24-25, 26-28, 48-50, 90-93, 108-111, 150-151, 170-172, 268-270, 375-376, 368-370).
This revised description fits better with our results and our discussion.

For the issue concerning the sensitivity of the individual \$^{231}\text{Pa}/^{230}\text{Th}\$ records to capture AMOC variations,
we further refer to Fig. 3. Here, we have quantified to what an extent our data is capable of capturing

hypothetical changes in AMOC strength. As a function of bioturbation depth and residence time of ^{231}Pa
and ^{230}Th , and depending on the amplitude of AMOC perturbation, the duration of detectable variations
differs from site to site as a result of core location and water depth (lines 187-207). We also address this
in the conclusion (lines 387-390). To clarify our model approach, we have significantly revised the
explanation in the methodology section (lines 513-564).

R1.2
Line 27: provide an estimate of the time resolution
We removed the vague term 'high-resolution' and now refer here to the millennial resolution of the
individual sedimentary $^{231}\text{Pa}/^{230}\text{Th}$ records, now provided in Supplementary Table 1 (lines 26-28).

R1.3
Line 28-29: this sentence is unclear
We agree, this sentence was indeed unclear. We now revised it to (lines 28-31):
"We found that during the Early Holocene the AMOC recovered from its weak deglacial state. At around
6.5 ka BP, the AMOC reached its pre-industrial strength (18.5 ± 1.5 Sv) and remained constant until
industrial times. The only prominent weaker AMOC state occurred between 9-8 ka BP (16 ± 1.3 Sv)."

With regard to R1.2 and R1.3, we have made several more changes to the abstract to address the issues
and ensure compliance with the formatting guidelines (lines 22-33).

R1.4
Line 57-58: This sentence is wrong, there exists (at least) one quantitative estimate of the AMOC over a
large part of the Holocene in Jomelli et al. (2022).
Jomelli et al. (2022) reconstruct glacier extents during the Holocene. For this, they made also use of a
semi-empirical model to estimate variations in the AMOC strength as a potential driver of glacier
growth/decrease. These estimates were based on proxies/observations of SST and sortable silt (Ayache
et al., 2018; Caesar et al., 2021; Caesar et al., 2018; Kaufman et al., 2020; Thornalley et al., 2013) which
have been implemented in a linear model and LOVECLIM simulations. This is a remarkable approach for
a paper dealing in first line with retreats and advances of glaciers.
We have now rephrased this sentence more carefully and refer to Jomelli et al. (2022) and Colin et al.
(2019) as additional sources of Holocene AMOC estimates (lines 55-57).

R1.5
Line 60: "fundamental principle" is a bit strong. This is an assumption and has nothing to do with a
fundamental physical principle. This proxy, as much of others, is still debated in terms of exact
representativity.

With the words 'fundamental principle' we intended not to describe a natural law, but to summarize
the idea behind the proxy. We agree that this can be understood as a too strong statement regarding
the assumptions made for the $^{231}\text{Pa}/^{230}\text{Th}$ proxy. We have revised the sentences accordingly by changing
it to (lines 59-64):

" ^{231}Pa and ^{230}Th are both radioactive decay products of ^{235}U and ^{238}U , respectively, which are well-mixed
in ocean water, resulting in the assumption of a spatiotemporally constant production rate of 0.093
[23]. Bound to particles, ^{231}Pa and ^{230}Th are removed from the ocean and are eventually buried in
bottom sediments. The $^{231}\text{Pa}/^{230}\text{Th}$ proxy is based on the differing removal rates, with the more particle-
reactive ^{230}Th being adsorbed more intensely than ^{231}Pa ."

R1.6

Line 71-72: Is there any instrumental validation of this proxy?

Yes, there is observational evidence from seawater measurements within the GEOTRACES program
(Deng et al., 2018; Deng et al., 2014; Kretschmer et al., 2008) that affirm the advective transport of ^{231}Pa
over ^{230}Th with the AMOC, causing a relative deficit of ^{231}Pa in the (North) Atlantic Ocean. To make this
point clearer we now refer in the manuscript to the manifold (conceptual to complex) models of various
complexities, that suggest a strong connection between AMOC and $^{231}\text{Pa}/^{230}\text{Th}$ in the open ocean (Gu
and Liu, 2017; Marchal et al., 2000; Rempfer et al., 2017; van Hulst et al., 2018). We revised it to (lines
69-72):

“This simplified conceptual view thus suggests that during times of weaker AMOC less ^{231}Pa would be
advected southwards and a sediment core in the North Atlantic would hence record higher $^{231}\text{Pa}/^{230}\text{Th}$
values as maintained from model approaches of different complexities [27-31].”

R1.7

Line 108: “Higher than 0.5 ka” is not very precise. Lower than?

Line 123: “centennial scale resolution” is not allowing to assess centennial variability (cf. Fig. R1)

In the process of revising our statements on temporal resolution (addressed in R1.1 & R1.2) we have
changed all parts in the manuscript related to temporal resolution.

Here, we now refer to the newly added Supplementary Table 1 to get the mean temporal $^{231}\text{Pa}/^{230}\text{Th}$
resolution of every core (lines 90-95).

R1.8

Line 125-126: it might be nice to use pseudo-proxy approach to evaluate the representativity of the
sampling in terms of AMOC reconstruction (e.g. Ayache et al. 2018). This means using a model modelling
this proxy and see if the sampling is permitting to correctly represents the AMOC. It is unclear if this
has been done in former publications. I doubt so for this specific sampling. This is a necessary validation
test to have any confidence in this reconstruction.

We thank the Reviewer#1 for bringing up this point.

We assessed the sensitivity of $^{231}\text{Pa}/^{230}\text{Th}$ to AMOC changes under pre-industrial (PI) boundary
conditions by using 20 idealized simulations performed with the Bern3D model. The AMOC was
perturbed with varying freshwater inputs in the North Atlantic, creating a range of AMOC strengths and
corresponding $^{231}\text{Pa}/^{230}\text{Th}$ values. To visualize the modelled AMOC responses to these freshwater
inputs, we now added Supplementary Fig. 7. For each core location, we extracted $^{231}\text{Pa}/^{230}\text{Th}$ data from
the model, applied low-pass filtering, and generated bioturbated pseudo-proxy records
(Supplementary Fig. 6). These were normalized to PI values ($\Delta^{231}\text{Pa}/^{230}\text{Th}$) and fitted to AMOC strength
using polynomial regression. The resulting site-specific relationships, including their uncertainties, are
now provided in Supplementary Table 2.

To summarize, we generated artificial $^{231}\text{Pa}/^{230}\text{Th}$ proxy data by simulating various artificial AMOC
changes. Based on this approach, we consider our methodology to align with a pseudo-proxy
framework. However, we previously did not call it a pseudo-proxy approach. We now revised the
discussion (lines 216-221), methods (lines 519-533) and the Supplement (Supp. Fig. 6-8) on how we
generated our pseudo-proxy records, and improved the explanation.

R1.9

Line 159: a pseudo-proxy approach might be also a way to evaluate the type of variability you are able
to reproduce from your temporal sampling. Which temporal sampling in a model do you need to be able
to reproduce centennial variability from a given model (e.g. Jiang et al. 2021)

A pseudo-proxy approach was used by low-pass filtering and subsampling the simulated $^{231}\text{Pa}/^{230}\text{Th}$
evolutions. Our pseudo-proxy approach assumes a bioturbation length of 5 cm, which depending on

the sedimentation rate, translates to low-pass filtering simulated $^{231}\text{Pa}/^{230}\text{Th}$ evolutions with a cut-off
ranging from 220 years to 560 years. Then, the “bioturbated” record was sub-sampled assuming, for
simplicity, a constant sampling rate of 1 cm (60-112 years). The Methods section has been expanded to
include an explanation on how we generated these pseudo-proxy $^{231}\text{Pa}/^{230}\text{Th}$ records (lines 519-533).
For more details please refer to R1.8.

R1.10
Line 179-180: on Fig. 4, one can clearly see that the reconstruction is far flatter than the original proxy
records, which were showing some substantial variability, which is totally smoothed by the model and
the reconstruction procedure.

As a first step in assessing the Holocene variability we detrended the sedimentary $^{231}\text{Pa}/^{230}\text{Th}$ records
to account for post-deglacial effects (Table 1). We added the missing abbreviation of ‘detrended’ (det.)
to the caption of Table 1. Overall, we observe a relative standard deviation (RSD) of less than 10% for
each record, with four out of five records showing an RSD around 5%.

Although the $^{231}\text{Pa}/^{230}\text{Th}$ variations during the Holocene appear significant in Fig. 2, the statistical
analysis here applied reveals that the internal variability of the individual $^{231}\text{Pa}/^{230}\text{Th}$ records during the
Holocene is quite small (below 10%).

Even when considering the summarized analytical uncertainties of observations and the model
approach, most of the observed changes remain within this margin of error. This becomes particularly
evident when comparing the Holocene $^{231}\text{Pa}/^{230}\text{Th}$ records to periods of more substantial AMOC
changes, such as the YD and HS1 (Ng et al., 2018).

We believe that the strength of our approach lies in the ability to combine multiple individual $^{231}\text{Pa}/^{230}\text{Th}$
down-core records into one record of AMOC strength. Due to this we are cautious in interpreting
smaller $^{231}\text{Pa}/^{230}\text{Th}$ variations present in individual records (see also 4.2 event at Bermuda Rise, Fig. 5)
especially given the analytical complexity of the applied proxy.

In this statistical and analytical context, we believe that the GAM fit represents the most robust
statistical approach for identifying the overarching trends from these different records. We revised the
description of the GAM calculation and its error propagation in the Discussion (lines 229-235), Methods
(lines 555-564) and now additionally refer to Wood et al. (2017) as a reference of our GAM approach in
the revised manuscript (lines 560-561).

R1.11
Line 193-194: what is the internal variability of this model? does it have any multi- centennial variability
when run in preindustrial conditions? If not, it is very likely that it might not be able to reconstruct
centennial variability, which can be found in a number of CMIP6 models (e.g. Bonnet et al. 2021). Also
what are the biases for present-day water masses representation? This might strongly affect the
capability of reconstruction the AMOC in the past based on depth of water masses. For instance, it is
already difficult to reconstruct the AMOC in data assimilation system before the Argo period, because
the observation sampling and the biases in the model are too strong, cf. Karspeck et al. (2015)

The Bern3D model does not exhibit any self-sustained AMOC oscillations on (multi-)centennial
timescales under pre-industrial boundary conditions. However, we want to emphasize that this has no
implications for the interpretation of the proxy-enabled simulations of the present study. Here, we force
the model by artificial freshwater fluxes to the North Atlantic to induce different AMOC responses. We
are here solely interested in the relationship between the simulated AMOC variability (even though it is
forced) to the directly simulated proxy evolution in different locations of the North Atlantic where the
investigated sediment cores are situated. Our approach is therefore different from the one by Karspeck
et al. (2015). More information on the freshwater forcing can be found in replies R1.13 and R2.13, as
well as in the revised Method section of the manuscript (lines 511-533) and Supplementary Fig. 7.

Model biases for the pre-industrial state are described in detail in Pöppelmeier et al. (2023b). Briefly,
similar to most Earth systems models, the Bern3D model exhibits a too shallow North Atlantic Deep
Water (NADW) cell, as Nordic Seas overflow waters are entrained at too shallow depths. As such, the
deepest parts of the North Atlantic are too old compared to observations and exhibit a too large fraction
of southern-sourced waters. Consequently, Antarctic Intermediate Water is also too shallow as it is
pushed upward by too shallow NADW. We now refer to the study by Pöppelmeier et al. (2023b) in the
Method section (lines 516-517). While this introduces some biases in the model assessment of past
AMOC variability, water mass changes are not directly influencing the $^{231}\text{Pa}/^{230}\text{Th}$ proxy, which captures
primarily the regional-scale advection.

We captured the range of model outputs and some of its uncertainty by performing a number of
sensitivity experiments that together determine the sensitivity of $^{231}\text{Pa}/^{230}\text{Th}$ to AMOC changes at a
specific location (Supplementary Fig. 6). We are also now giving the simulated site-specific $^{231}\text{Pa}/^{230}\text{Th}$ to
AMOC relationships and its corresponding uncertainty in Supplementary Table 2.

R1.12

Line 221: “modest (-4,9 Sv) AMOC weakening”. I do not think such a weakening over 100 years is modest!
It is larger than what is projected in some CMIP6 models for 2100 and represents more than 25% of
present-day AMOC! As a point of comparison, Jomelli et al. (2022) estimated that mid-Holocene might
be possibly 3- 4 Sv larger than present-day. Thus, the method and proxy records used in this paper does
not allow to contradict results from this former AMOC reconstruction. What this approach is allowing to
conclude is just that there were no DO-like events during the Holocene, which is already well known!

We agree that, relative to the AMOC evolution during the Holocene warm period, and in the light of
potential future changes in AMOC strength, a 4.9 Sv reduction can indeed not be considered to be
‘modest’. We have hence removed ‘modest’ from this sentence (lines 202-204).

Regarding the mid-Holocene AMOC maximum:

Jomelli et al. (2022) states: “The difference in AMOC strength between the mid-Holocene (6 ka) and the
preindustrial period is 3–4 Sv, ...”.

Both, the reconstructed AMOC strength records of this study and by Jomelli et al. (2022) (Fig. S6 in
Jomelli et al. (2022)) share the perception of the strongest Holocene AMOC around 7 ka BP. We are
therefore able to detect the same signal. We thus do not see a significant contradiction to these previous
results, besides our study reveals a smaller amplitude difference between PI and mid-Holocene when
compared to Jomelli et al. (2022) , with a maximum of ~19 Sv at ~7ka BP, an issue we are addressing in
lines 263-270. Our approach is based on the results of a kinematic proxy and are therefore independent
from SST-based reconstructions (Ayache et al., 2018; Jomelli et al., 2022). We thus deliver an additional,
but independent way to estimate past AMOC strength.

R1.13

Line 278: It is useful to keep in mind that response of climate models to a given amount of freshwater is
very model dependent (cf. Stouffer et al. 2006, and still true in most recent models, cf. Jackson et al.
2023).

We fully agree with the Reviewer#1 that climate models have a wide range of sensitivities to freshwater
forcing and to the location where it is applied. However, the aim of this study is not to reconstruct
potential freshwater inputs into the North Atlantic that might have disrupted the AMOC during the
Holocene nor to directly simulated AMOC variability during the Holocene. Instead, we simulated
idealized scenarios of AMOC variability with the proxy enabled model to assess the relationship
between AMOC and $^{231}\text{Pa}/^{230}\text{Th}$ (pseudo-proxy). We then apply this relationship to the reconstructed
$^{231}\text{Pa}/^{230}\text{Th}$ timeseries to get an estimate of what their variability (or lack thereof) implies for past AMOC

changes and what signal, in terms of length and magnitude of perturbation, could be detected with this
method (lines 187-207) (see also R2.13).

R1.14

Line 302: Your reconstruction is not allowing to exclude the occurrence of a very short-scale events
of few decades, related for instance only to a subpolar gyre instability (e.g Sgubin et al. 2019).

Yes, due to the limited temporal resolution of the $^{231}\text{Pa}/^{230}\text{Th}$ proxy data we would not be able to
detect AMOC weakening events that lasted a few decades, such as subpolar gyre instabilities. We
extensively address the problem of the impossibility of detecting very short events (like the
subpolar gyre instability) by our approach in lines 202-207, 387-390 and as depicted in Fig. 3. We
now revised the explanation of this approach in the Methods (lines 519-533). The approach of our
revised work allows to evaluate and quantify millennial to sub-millennial AMOC variability in the
North Atlantic. Decadal changes are therefore not within the scope of this work.

However, the context of the sentence referred to by Reviewer#1 (lines 299-301), pertains not to
our findings but to those of Bond et al. (2001). In this sentence as well as the previous sentences,
we summarize the evidences from various studies, proxies, and methodologies to provide a more
comprehensive understanding of the AMOC during the 4.2 ka BP event.

R1.15

Line 315-320: In the higher resolution AMOC reconstruction from Ayache et al. (2018), there is an
AMOC fluctuation around 4.2 ka. Can it be possible that it is not detected here due to low sensitivity
to AMOC changes of the method used?

While our Holocene AMOC reconstruction shares many similarities with that of Ayache et al. (2018),
particularly during the Early to Mid-Holocene (10-6 ka BP), these similarities lessen in the Late
Holocene (see also R1.12). Ayache et al. (2018) reports an AMOC increase lasting about 200-250 years
during the 4.2 ka BP event. However, our data do not show any such changes and stay virtually
constant over this time period. While the duration of this event (longer than 200 years) could be
sufficient for $^{231}\text{Pa}/^{230}\text{Th}$ to detect an AMOC shift (Fig. 3), the event's proposed magnitude could be
too subtle. This notable discrepancy may result from factors such as the low sedimentation rates in
the deep ocean, bioturbation, and the residence times of ^{231}Pa and ^{230}Th , limiting the method's ability
to detect small/short AMOC perturbations. We extensively addressed the limitations of this approach
in the discussion (lines 202-207, 263-270), including Fig. 3 and also in the conclusion (lines 387-390).

While our $^{231}\text{Pa}/^{230}\text{Th}$ records rule out a 200-250 yrs AMOC shift of >4.9 Sv during the 4.2 ka BP event,
we cannot entirely dismiss the possibility of a shorter (<180 yrs) and weaker (<4.9 Sv) AMOC change.
However, the climatic effects of a potential uniform AMOC increase during the 4.2 ka BP event, as
suggested by Ayache et al. (2018) and primarily inferred from SST data, are not supported by more
recent studies (Bradley and Bakke, 2019; McKay et al., 2024; Yan and Liu, 2019).

McKay et al. (2024) reports only minimal and non-uniform climatic changes in the North Atlantic
during the 4.2 ka BP event. This is also supported by Bradley and Bakke (2019): "*...the absence of a
strong signal of an abrupt climatic event at 4.2 ka BP suggests that... - it is unlikely that the North
Atlantic Ocean circulation played a driving role*".

Our main message in this chapter is, that besides the paleoclimatic evidences mentioned above, our
results of the kinematic $^{231}\text{Pa}/^{230}\text{Th}$ proxy also indicate only minimal AMOC variability during the 4.2
325 ka BP event. Thus, changes in AMOC strength are unlikely be the driving factor behind the 4.2 ka BP
climatic event. To emphasize the consistency of our AMOC results with the latest global findings on
the 4.2 ka BP event, we now refer to McKay et al. (2024) as an additional reference (lines 334-336).

R1.16

Line 348: How is the uncertainty of 1.1 Sv cited in parenthesis is computed? It seems largely
underestimated in my view. Same remark with the number 1.3 Sv line 180.

For the quantitative estimate of the AMOC (Fig. 4) we first converted the sedimentary $^{231}\text{Pa}/^{230}\text{Th}$ data
into Sverdrup (Sv), by using our pseudo-proxy modelling approach. We accordingly converted the
respective $^{231}\text{Pa}/^{230}\text{Th}$ uncertainties into Sv and also included the uncertainties of the simulations for
the respective core locations. Next, we implemented a Generalized Additive Model (GAM) fit, and also
provided its mean 95% confidence interval (CI). This fit incorporated the weighting of the previously
explained individual analytical/observational uncertainties ($1/\text{error}^2$). These results of the GAM fit with
CI are given in the Supplementary Data sheet.

We agree with the Reviewer#1, that the uncertainty of the GAM fit appears quite low. In contrast, the
individual data points exhibit larger errors, typically ranging from 0.5 to 2.5 Sv. However, since we only
interpret the results of the GAM fit in the paleocirculation context, we included its 95% CI.

We streamlined the explanation of our approach in the Discussion (lines 229-235) and expanded on the
explanation in the Methods (lines 555-564). In this Methods section, we also expanded on the
calculation of the GAM, its uncertainties, the error propagation, and included the reference of the
statistical GAM approach.

**Reviewer #2**

Gerber et al. investigates variability in Atlantic Meridional Overturning Circulation (AMOC) during the
Holocene (last 11.7 ka) from two entirely new sedimentary $^{231}\text{Pa}/^{230}\text{Th}$ records from the North Atlantic,
four existing mid-latitude records, and using Bern3D, an Earth System Model of intermediate complexity.
This is an important topic, as the AMOC has been identified as a core climate tipping element in the Earth
system (e.g. Armstrong McKay et al. 2022, Science); IPCC AR6 has stated it is very likely the AMOC will
decline over the 21st century, but there is low confidence in the quantitative projections due to poor
process understanding (Fox-Kemper et al., 2021).

The new proxy reconstructions presented in this study address a highly uncertain, but important topic
with respect to the global climate. This work could be a useful contribution to Nature Communications.
The main findings are that fluctuations in AMOC strength were highest in the early Holocene as the ice
sheets deglaciated, with a pronounced slowdown during the 8.2 ka BP meltwater pulse event, and
centennial-scale AMOC variability remained low since 6.5 ka BP, with no notable change at 4.2 ka BP.
However, I note that these results are model-dependent because the interpretation of the records, and
in particular the differences between the various records, is based on simulations with one intermediate
complexity model. I have some concerns with respect to the modelling aspect of this work that I hope
the authors can address. This review is focused on the Earth system modelling as the proxy
reconstruction is outside of my particular expertise.

We sincerely thank Reviewer#2 for their review and constructive feedback on our work. We appreciate
insightful comments regarding our simulations with the Bern3D model.

For this point-by-pointy response, we chose to split up some comments to allow for a clearer and more
detailed response.

R2.1

The experimental design for the idealised Bern3D simulations make sense, but the methods description
is vague and lacking detail. As the methods are currently written, I am convinced I would not be able to
set up and run these same experiments without more information, so the authors should consider the
reproducibility of these simulations.

We regret that the information on the model runs and the model itself given in the manuscript were not
sufficient to allow a full assessment of the modelling approach. We have revised and extended the
description of the model in the updated version of the manuscript.

Our idealized \$^{231}\text{Pa}/^{230}\text{Th}\$ sensitivity simulations were following a pseudo-proxy approach (see R1.8).
We also expanded the Methods section with a detailed explanation on how we generated the pseudo-
proxy records (lines 535-564). See also our response to R1.8.

We additionally added two new figures to the Supplement, showing the freshwater forcing its
corresponding AMOC strength (Supplement Fig. 7) and the Bern3D AMOC stream functions during the
PI, weakest and strongest states (Supplementary Fig. 8). The caption of Supplementary Fig. 6 has been
revised and expanded on.

(lines 535-553) "Next, we assessed the sensitivity of \$^{231}\text{Pa}/^{230}\text{Th}\$ to AMOC changes, under PI boundary
conditions, with a suite of 20 idealized simulations (Supplementary Fig. 6). In each simulation, the AMOC
is forced to a rapid increase in strength and gradual slowdown of different lengths (500 – 2000 years) and
amplitudes (4 – 23 Sv) by applying varying freshwater perturbations in a latitudinal band between 45°N
and 70°N in the North Atlantic (Supplementary Fig. 7). For verification, the stream function of the AMOC
for the PI state, as well as for the strongest and weakest states, has been included in Supplementary Fig.
8. Thereby, we obtain a wide range of AMOC strengths and corresponding \$^{231}\text{Pa}/^{230}\text{Th}\$ values."

R2.2

As some examples for more detail: What was the spin-up procedure?

See also R1.8. and R2.1.

We have expanded the methods section to include a more detailed overview on the spin-up procedure
and simulation set-up.

We added the following sentence (lines 521-524): “First, the model was spun up for 45000 years under
PI conditions until an equilibrium state was reached. Greenhouse gas concentrations are set to CO₂ =
278.05 ppm, CH₄ = 721.89 ppm, N₂O = 272.96 ppm and ²³¹Pa/²³⁰Th tracers are only included after 35,150
401 years of spin up.”

R2.3

How was the model validated?

The ocean physics were tuned to match modern World Ocean Atlas fields of tracer distributions as
described in Pöppelmeier et al. (2023b). The ²³¹Pa/²³⁰Th tracers were first implemented by Rempfer et
al. (2017) and further developed and re-tuned by Pöppelmeier et al. (2023a). For this, scavenging
parameters and desorption constants were tuned to match modern observations of dissolved Pa and Th
concentrations from the GEOTRACES database based on a Latin hypercube sampling approach of the full
parameter space. We now briefly mention this in the Method section (lines 511-517).

R2.4

What volumes of freshwater input were applied in each of the idealised simulations (AMOC weakening
of 4.8, 6.8, and 8.5 Sv), and what were these amounts based on?

It is important to note, that we do not use the Bern3D model to simulate the Holocene AMOC. Instead, we
use the model to examine the dependency of ²³¹Pa/²³⁰Th as a function of AMOC strength at the locations
of our cores, by employing various AMOC strength scenarios. The regional ²³¹Pa/²³⁰Th sensitivity is then
used for reconstructing the Holocene AMOC strength from the actual ²³¹Pa/²³⁰Th observations. Freshwater
fluxes were used to force the distinct AMOC states in the model. The freshwater fluxes are time dependent
for the generation of pseudo-proxy records and are plotted in the new Supplement Fig. 7.

The freshwater fluxes resulting in these different AMOC weakening events were 0.04, 0.082, and 0.15
422 Sv. The amounts were solely chosen to generate 3 different AMOC states and are not based on
observational constraints of freshwater fluxes. We now mention this as well in the Methods section (lines
525-533).

R2.5

The authors should do more to demonstrate that their proxy interpretation is not dependent on this
particular intermediate complexity model, or at least better contextualise their results noting the
diversity of AMOC representation in models of various levels of complexity. Highlighting the modelling
challenge, Weijer et al. (2020) show the wide range of AMOC strength and response to SSP scenarios in
CMIP6 models; He and Clark (2022) demonstrate that some AOGCMs may be too sensitive to freshwater
input; Obase et al. (preprint) show substantial differences in AMOC strength among PMIP4 models over
the last deglaciation. In this study, the idealised simulations are used to investigate the detectability of
AMOC perturbations at the individual sites. The main question I'd like the authors to consider here is:
How would the proxy interpretation change if using a different ESM?

We have determined the correlation between the AMOC and sedimentary ²³¹Pa/²³⁰Th for each core
location using the simulations performed with the Bern3D model (Supplementary Fig. 6). This relationship
between the AMOC and ²³¹Pa/²³⁰Th is broadly consistent with results from other Earth System Models (Gu
and Liu, 2017; Gu et al., 2024; van Hulten et al., 2018), leading to qualitative agreement in the observed
trends.

However, it remains possible that a different Earth System Model could yield a different slope for the
AMOC–²³¹Pa/²³⁰Th relationship. While this might result in small differences in the absolute Sverdrup
values, the consistency of the signal — as observed in sedimentary ²³¹Pa/²³⁰Th records (Fig. 2 and Table 1)
— would remain largely unchanged. Still, we fully agree with the reviewer that performing the same
simulations with different Earth system models would provide a more robust uncertainty assessment. Yet,
to this day only a few models have Pa and Th tracers explicitly implemented in addition to having the
capability to simulate multi-millennial timescales.

To address this point, we have reworked the Methods section and included all necessary details of our
simulations performed with the Bern3D model (lines 511-564). Thus, other groups will be able to
reproduce our simulations and utilize the sedimentary ²³¹Pa/²³⁰Th data (Supplementary Data Sheet) to
replicate our results of Fig. 4. Additionally, we have added Supplementary Table 2 to the revised version
of the manuscript, providing the site-specific AMOC-to-²³¹Pa/²³⁰Th relationships used for calibrating the
sedimentary ²³¹Pa/²³⁰Th in this study. These relationships may serve as a reference for comparison with
future simulations.

We further emphasize that our results do not depend on the freshwater sensitivity of the employed model.
We used the freshwater forcing only to induce AMOC changes that can be related to the directly simulated
²³¹Pa/²³⁰Th. As such, we solely utilize these simulations to estimate the relationship between AMOC and
²³¹Pa/²³⁰Th changes, without inferring anything about real past freshwater fluxes.

R2.6

Line 1: Should the title be “multi-centennial”? Just noting the temporal resolution of the records is 0.5 ka.
We thank Reviewer#2 for bringing up this issue.

As we already mentioned in the introduction, the description of the temporal resolution is one of the
main topics brought up in this review and is particularly emphasized by Reviewer#1. Therefore, we would
like to respectfully refer to points R1.1 and R1.2, where we have discussed this topic in detail.
Nevertheless, we would like to briefly summarize that we have now revised the manuscript (lines 1, 24-
25, 26-28, 48-50, 90-93, 108-111, 150-151, 170-172, 268-270, 375-376, 368-370) and the title (line 1) by
calling the temporal resolution of our individual ²³¹Pa/²³⁰Th records ‘millennial’ and the resolution of the
Gam fit ‘sub-millennial’.

R2.7

Line 73-105: I find this paragraph difficult to follow because it covers so many topics: how the
²³¹Pa/²³⁰Th has been used previously; 8.2 and 4.2 ka BP events; sedimentological conditions at
particular sites. I recommend revising for brevity and clarity.

We agree that this paragraph was not clear enough and contained redundancies that are discussed
elsewhere in the manuscript. We have, thus, shortened and streamlined it, as suggested by the
Reviewer#2 (lines 83-89).

R2.8

Line 90: Northern Hemisphere
Corrected (line 84)

R2.9

Line 92: Noting two “..”
Corrected (line 87)

R2.10

Figure 1: Specify which records are “new”

To provide a brief overview of the new data from our study, we have revised the manuscript (lines 108-
111) and the caption of Fig. 1 accordingly, that now reads (lines 100-103):

“New $^{231}\text{Pa}/^{230}\text{Th}$ records: ODP 983 and GeoB18529-2. $^{231}\text{Pa}/^{230}\text{Th}$ records with new additional data: ODP
1063 and ODP 1059. Unchanged previously published $^{231}\text{Pa}/^{230}\text{Th}$ records: OCE326-GGC5 and KN140-2-51
GGC.”

Additional information about the new $^{231}\text{Pa}/^{230}\text{Th}$ data (lines 398-434), as well as updates on the age
models (lines 437-454) can be found in the Method section and in the Supplementary Data Sheet.

R.2.11

Figure 2: Triangle colours need explaining. Are there no age model tie points for ODP 983?

We have revised the caption of Fig. 2 to clearly indicate that the triangles are color-coded with the
corresponding $^{231}\text{Pa}/^{230}\text{Th}$ records (lines 143-146).

In this regard we also corrected the edge color of the empty/not-filled triangles in Fig. 2. Their edge was
previously black but should have been brown, as these empty age tie points belong to the age model of
ODP 1063. These empty age tie points correspond to previously published ^{14}C data that were incorporated
into the calculation of the updated age model of this study (see Supplementary Data Sheet).

We used the previously published age model for core ODP 983 (Barker, 2021; Waelbroeck et al., 2019)
without applying any changes (lines 437-438). To avoid further complicating this already crowded figure,
we have decided to only included ^{14}C age model tie points from age models that we generated or revised
an already existing one.

R2.12

Line 185: Double check grammar of this sentence.

Revised (lines 172-174)

R2.13

Line 517: Do you consider the impact of freshwater flux in the Southern Ocean (e.g. Bakker et al. 2017)? I
recommend performing additional sensitivity experiments with Bern3D to investigate how site
detectability changes with a Southern Hemisphere-sourced AMOC perturbation.

For the simulations performed here freshwater was only applied to the North Atlantic. Importantly, these
freshwater forcings were not aimed at generating realistic AMOC responses for the Holocene. Instead, we
use them only to estimate the relationship between AMOC and $^{231}\text{Pa}/^{230}\text{Th}$ changes. We only consider the
first order effects and hence the drivers of the AMOC change are not the objective of our study. Although
the relationship between simulated AMOC and $^{231}\text{Pa}/^{230}\text{Th}$ could be different if one would apply freshwater
hosing to the Southern Ocean, the general first-order response should remain the same, as $^{231}\text{Pa}/^{230}\text{Th}$
does not trace water mass origins but regional-scale advection. Further, from our experience applying
freshwater hosing to the Southern Ocean only generates a small change in AMOC. We therefore
respectfully refrain from performing additional sensitivity experiments.

R2.14

Supplementary Information: I recommend adding a figure showing the streamfunction fields for the
Preindustrial control and the AMOC perturbation experiments (e.g. see Boulton et al., 2014, Nature
Comms., Fig 1).

Thank you for the suggestion. We have added Supplementary Fig. 8, illustrating the AMOC stream function
during the pre-industrial control state, as well as during the strongest and weakest AMOC phases. The new
figure is mentioned in the Method section (lines 590-593).

**Reviewer #3**

Gerber and coauthors analyzed multiple high-resolution $^{231}\text{Pa}/^{230}\text{Th}$ sedimentary records of the North
Atlantic. The authors also used climate model simulations to quantify past AMOC strength by addressing
the different depths and locations of sedimentary records. They estimated the strength of AMOC during
the Holocene, particularly for the minimum value of the Holocene and trends.

The materials and methods seem good, but I still don't understand some points related to climate model
simulations, which is important in estimating quantitative AMOC strength changes. I hope the revised
manuscript clarifies the following points.

We sincerely thank Reviewer#3 for their review and constructive feedback on our work. We appreciate
insightful comments regarding the description of our simulations with the Bern3D model.

R3.1

L195-202: These lines correspond to explanations of Figure 3 and Supplementary Figure 3, which makes
sense. However, the method section says the 20 idealized simulations with different lengths and
amplitudes by applying varying freshwater perturbations for the Bern3D model (L514-519). Figure 3
shows that the minimum values of the AMOC in seven rows are identical (-4.9, -6.8, -8.5Sv), so I assume
the 20 simulations are scaled to create 21 scenarios but not sure. Please explain how to get 21 AMOC
time series in Figs.3 and S3 from the original 20 Bern3D model simulations.

We apologize for the confusion. We have now clarified this issue.

We revised the brief explanation in the Discussion on our modelling approaches (lines 187-195, 216-235)
and completely reworked the detailed explanation in the Methods (lines 499-564).

We clarified, that we performed to sperate sets of simulations. The first set consist of 21 simulations (three
different AMOC weakenings of 4.9, 6.8, and 8.5 Sv and perturbation lengths of 100, 120, 140, 160, 180,
200, and 300 years) (Fig. 3) (lines 519-533). This set was previously briefly described in the main text, but
not in the Method section.

The second set of simulations consist of 20 experiments described in the Method section and was used to
derive the AMOC- \$^{231}\text{Pa}/^{230}\text{Th}\$ relationship (lines 535-564) depicted in Supplementary Fig. 6.

R3.2

Figure 6 needs the derivation of individual blue dots and methods to get dashed red lines. And I couldn't
find the reference to Figure 6 in the manuscript.

We have moved the previous Fig. 6 to the Supplement (now Supplementary Fig. 6) and expanded on its
caption. We additionally added more explanation on how we obtained the pseudo-proxy records displayed
in Supplementary Fig. 6 to the Methods (lines 535-553). We also included the reference of Supplementary
Fig. 6 in the manuscript (lines 536, 551).

**References**

- Ayache, M., Swingedouw, D., Mary, Y., Eynaud, F., and Colin, C., 2018, Multi-centennial variability of the
AMOC over the Holocene: A new reconstruction based on multiple proxy-derived SST records:
Global and Planetary Change, v. 170, p. 172-189.
- Barker, S., 2021, Planktic foraminiferal and Ice Rafted Debris (IRD) counts from ODP Site 983, PANGAEA.
- Bond, G., Kromer, B., Beer, J., Muscheler, R., Evans, M. N., Showers, W., Hoffmann, S., Lotti-Bond, R.,
Hajdas, I., and Bonani, G., 2001, Persistent Solar Influence on North Atlantic Climate During the
Holocene: Science, v. 294, no. 5549, p. 2130-2136.
- Bradley, R. S., and Bakke, J., 2019, Is there evidence for a 4.2-ka BP event in the northern North Atlantic
region?: Climate of the Past, v. 15, no. 5, p. 1665-1676.
- Caesar, L., McCarthy, G. D., Thornalley, D. J. R., Cahill, N., and Rahmstorf, S., 2021, Current Atlantic
Meridional Overturning Circulation weakest in last millennium: Nature Geoscience.
- Caesar, L., Rahmstorf, S., Robinson, A., Feulner, G., and Saba, V., 2018, Observed fingerprint of a weakening
Atlantic Ocean overturning circulation: Nature, v. 556, no. 7700, p. 191-196.
- Colin, C., Tisnérat-Laborde, N., Mienis, F., Collart, T., Pons-Branchu, E., Dubois-Dauphin, Q., Frank, N.,
Dapoigny, A., Ayache, M., Swingedouw, D., Dutay, J.-C., Eynaud, F., Debret, M., Blamart, D., and
Douville, E., 2019, Millennial-scale variations of the Holocene North Atlantic mid-depth gyre
inferred from radiocarbon and neodymium isotopes in cold water corals: Quaternary Science
Reviews, v. 211, p. 93-106.
- Deng, F., Henderson, G. M., Castrillejo, M., and Perez, F. F., 2018, Evolution of ²³¹Pa and ²³⁰Th in overflow
waters of the North Atlantic: Biogeosciences, v. 2018, p. 1-24.
- Deng, F., Thomas, A. L., Rijkenberg, M. J. A., and Henderson, G. M., 2014, Controls on seawater ²³¹Pa,
²³⁰Th and ²³²Th concentrations along the flow paths of deep waters in the Southwest Atlantic:
Earth and Planetary Science Letters, v. 390, p. 93-102.
- Gu, S., and Liu, Z., 2017, ²³¹Pa and ²³⁰Th in the ocean model of the Community Earth System Model
(CESM1.3): Geoscientific Model Development, v. 10, no. 12, p. 4723-4742.
- Gu, S., Liu, Z., Ng, H. C., Lynch-Stieglitz, J., McManus, J. F., Spall, M., Jahn, A., He, C., Li, L., Yan, M., and Wu,
599 L., 2024, Open ocean convection drives enhanced eastern pathway of the Glacial Atlantic
Meridional Overturning Circulation: Proc Natl Acad Sci U S A, v. 121, no. 45, p. e2405051121.
- Jomelli, V., Swingedouw, D., Vuille, M., Favier, V., Goehring, B., Shakun, J., Braucher, R., Schimmelpfennig,
I., Menviel, L., Rabatel, A., Martin, L. C. P., Blard, P. H., Condom, T., Lupker, M., Christl, M., He, Z.,
Verfaillie, D., Gorin, A., Aumaitre, G., Bourles, D. L., and Keddadouche, K., 2022, In-phase
millennial-scale glacier changes in the tropics and North Atlantic regions during the Holocene: Nat
Commun, v. 13, no. 1, p. 1419.
- Karspeck, A. R., Stammer, D., Köhl, A., Danabasoglu, G., Balsaseda, M., Smith, D. M., Fujii, Y., Zhang, S.,
Giese, B., Tsujino, H., and Rosati, A., 2015, Comparison of the Atlantic meridional overturning
circulation between 1960 and 2007 in six ocean reanalysis products: Climate Dynamics, p. 1-26.
- Kaufman, D., McKay, N., Routson, C., Erb, M., Davis, B., Heiri, O., Jaccard, S., Tierney, J., Datwyler, C.,
Axford, Y., Brussel, T., Cartapanis, O., Chase, B., Dawson, A., de Vernal, A., Engels, S., Jonkers, L.,
Marsicek, J., Moffa-Sanchez, P., Morrill, C., Orsi, A., Rehfeld, K., Saunders, K., Sommer, P. S.,
Thomas, E., Tonello, M., Toth, M., Vachula, R., Andreev, A., Bertrand, S., Biskaborn, B., Bringue,
613 M., Brooks, S., Caniupan, M., Chevalier, M., Cwynar, L., Emile-Geay, J., Fegyveresi, J., Feurdean, A.,
Finsinger, W., Fortin, M. C., Foster, L., Fox, M., Gajewski, K., Grosjean, M., Hausmann, S., Heinrichs,
615 M., Holmes, N., Ilyashuk, B., Ilyashuk, E., Juggins, S., Khider, D., Koinig, K., Langdon, P., Larocque-
616 Tobler, I., Li, J., Lotter, A., Luoto, T., Mackay, A., Magyari, E., Malevich, S., Mark, B., Massaferró, J.,
Montade, V., Nazarova, L., Novenko, E., Paril, P., Pearson, E., Peros, M., Pienitz, R., Plociennik, M.,
Porinchu, D., Potito, A., Rees, A., Reinemann, S., Roberts, S., Rolland, N., Salonen, S., Self, A., Seppa,
H., Shala, S., St-Jacques, J. M., Stenni, B., Strykh, L., Tarrats, P., Taylor, K., van den Bos, V., Velle,

G., Wahl, E., Walker, I., Wilmshurst, J., Zhang, E., and Zhilich, S., 2020, A global database of
Holocene paleotemperature records: *Sci Data*, v. 7, no. 1, p. 115.

Kretschmer, S., W. Geibert, Schnabel, C., Loeff, M. R. v. d., and Mollenhauer, G., 2008, Distribution of ^{230}Th ,
623 ^{10}Be and ^{231}Pa in Sediment Particle Classes: *Geochimica et Cosmochimica Acta*, v. 72, no.
Goldschmidt 2008, 12S.

Marchal, O., Francois, R., Stocker, T., and Joos, F., 2000, Ocean thermohaline circulation and sedimentary
$^{231}\text{Pa}/^{230}\text{Th}$ ratio: *Paleoceanography*, v. 15, p. 6.

McKay, N. P., Kaufman, D. S., Arcusa, S. H., Kolus, H. R., Edge, D. C., Erb, M. P., Hancock, C. L., Routson, C.
C., Zarczynski, M., Marshall, L. P., Roberts, G. K., and Telles, F., 2024, The 4.2 ka event is not
remarkable in the context of Holocene climate variability: *Nat Commun*, v. 15, no. 1, p. 6555.

Ng, H. C., Robinson, L. F., McManus, J. F., Mohamed, K. J., Jacobel, A. W., Ivanovic, R. F., Gregoire, L. J., and
Chen, T., 2018, Coherent deglacial changes in western Atlantic Ocean circulation: *Nature*
*Communications*, v. 9, no. 1, p. 2947.

Pöppelmeier, F., Jeltsch-Thömmes, A., Lippold, J., Joos, F., and Stocker, T. F., 2023a, Multi-proxy constraints
on Atlantic circulation dynamics since the last ice age: *Nature Geoscience*, v. 16, no. 4, p. 349-356.

Pöppelmeier, F., Joos, F., and Stocker, T. F., 2023b, The Coupled Ice Sheet–Earth System Model Bern3D
v3.0: *Journal of Climate*, v. 36, no. 21, p. 7563-7582.

Rempfer, J., Stocker, T. F., Joos, F., Lippold, J., and Jaccard, S. L., 2017, New insights into cycling of ^{231}Pa
and ^{230}Th in the Atlantic Ocean: *Earth and Planetary Science Letters*, v. 468, p. 27-37.

Thornalley, D., S. Barker, J. Becker, I. Hall, and Knorr, G., 2013, Abrupt changes in deep Atlantic circulation
during the transition to full glacial conditions: *Paleoceanography*, v. 28, p. 253-262.

van Hulst, M., Dutay, J. C., and Roy-Barman, M., 2018, A global scavenging and circulation ocean model
of thorium-230 and protactinium-231 with improved particle dynamics (NEMO–ProThorP 0.1):
*Geoscientific Model Development*, v. 11, no. 9, p. 3537-3556.

Waelbroeck, C., Lougheed, B. C., Vazquez Riveiros, N., Missiaen, L., Pedro, J., Dokken, T., Hajdas, I., Wacker,
645 L., Abbott, P., Dumoulin, J.-P., Thil, F., Eynaud, F., Rossignol, L., Fersi, W., Albuquerque, A. L., Arz,
H., Austin, W. E. N., Came, R., Carlson, A. E., Collins, J. A., Dennielou, B., Desprat, S., Dickson, A.,
Elliot, M., Farmer, C., Giraudeau, J., Gottschalk, J., Henderiks, J., Hughen, K., Jung, S., Knutz, P.,
Lebreiro, S., Lund, D. C., Lynch-Stieglitz, J., Malaizé, B., Marchitto, T., Martínez-Méndez, G.,
Mollenhauer, G., Naughton, F., Nave, S., Nürnberg, D., Oppo, D., Peck, V., Peeters, F. J. C., Penaud,
650 A., Portillo-Ramos, R. d. C., Repschläger, J., Roberts, J., Rühlemann, C., Salgueiro, E., Sanchez Goni,
651 M. F., Schönfeld, J., Scussolini, P., Skinner, L. C., Skonieczny, C., Thornalley, D., Toucanne, S., Rooij,
D. V., Vidal, L., Voelker, A. H. L., Wary, M., Weldeab, S., and Ziegler, M., 2019, Consistently dated
Atlantic sediment cores over the last 40 thousand years: *Scientific Data*, v. 6, no. 1, p. 165.

Wood, S. N., Pya, N., and Säfken, B., 2017, Smoothing Parameter and Model Selection for General Smooth
Models: *Journal of the American Statistical Association*, v. 111, no. 516, p. 1548-1563.

Yan, M., and Liu, J., 2019, Physical processes of cooling and mega-drought during the
4.2kaBP event: results from TraCE-21ka simulations: *Climate of the Past*, v. 15,
no. 1, p. 265-277.

**Authors response to reviews on the manuscript “Low variability of the Atlantic Meridional**
**Overturning Circulation throughout the Holocene” (NCOMMS-24-61663A)**

We sincerely thank the Reviewers and editor for their time and effort reviewing our manuscript.

Reviewer #4 raised an important issue regarding our IntCal20-based ^{14}C calibration and whether a
Marine20-based calibration might not be more appropriate. We regret any misunderstandings on this
topic, which we here aim to clarify by, addressing these concerns.

In the light of this, we have added a new section on the age model to the supplement, which includes more
detailed information transparently describing how we constructed the core chronologies.

Therefore, we believe that the valuable and helpful feedback further substantially improved the latest
version of the manuscript.

In the following, we present the Reviewers’ comments in black, with our responses provided in blue. Stated
line numbers refer to the revised versions of the manuscript and the Supplement without “tracked
changes”. Additionally, we are also providing a “tracked changes” version of both documents, so that it is
possible to follow all individual changes made by us in detail.

We hope that our revisions and clarifications will fulfill the Editors’ as well as the Reviewers’ requirements
and expectations.

**Reviewer #2**

This is the second round review of Gerber et al. "Low centennial-scale variability of the Atlantic
Meridional Overturning Circulation throughout the Holocene". To reiterate my previous review, this is
an important topic given that the AMOC has been identified as a core climate tipping element, but with
low confidence in quantitative projections. This record of Holocene AMOC variability is therefore an
important contribution.

In this revised manuscript, I appreciate the authors' efforts at addressing my previous concerns. The title
of the manuscript is now representative of the findings, the introduction is much improved, and I found
the revised methods section for Bern3D model very clear. The authors adequately addressed my concern
regarding freshwater forcing, clarifying that these experiments are idealised to show the effect of the
AMOC perturbation, but still noting the values, which allows for comparison to other models.

We again thank the Reviewer #2 for their thoughtful comments on our manuscript as well as the positive
assessment.

**Reviewer #3**
(no comments)

**Reviewer #4**

Review of "Low variability of the Atlantic Meridional Overturning Circulation throughout the Holocene"
Gerber et al Nature Communications

This manuscript presents Holocene reconstructions of AMOC generated with Pa/Th. Pa/Th is one of the
less intuitive proxies, so using a intermediate complexity earth system model really helps to interpret the
results. Despite the relative insensitivity of the proxy system to moderate changes in AMOC, I think the
conclusions that anthropogenic driven changes will be larger than anything in the Holocene is
reasonable.

The authors seem to have addressed concerns raised by the reviews in round one. Now that the ms
focuses on the millennial scale trends (the previous claim of centennial scale was unsupported), the
demands on the age-depth models are reduced. This is fortunate. The age-depth models are sufficient
for the current work, but not for claims of centennial scale.

R4.1

The ms reports that it improves the age-depth model for ODP 1063 by adding 5 extra dates but does not
show the resulting model so the reader cannot evaluate how reliable it is. I cannot tell what type of
model was used (and the the original paper for this core does not help). I don't understand why intcal20
was used - is this a typo - marine20 would seem to be a more appropriate calibration curve.

We thank Reviewer #4 for bringing up this point.

It is correct that using IntCal20 alone for marine samples would result in calibrated ages underestimating
the true age of the samples, as it does not account for the marine reservoir age.

We believe that the insufficient explanation of our radiocarbon age calibration in the previous
manuscript led to a misunderstanding of our approach. Our goal here is to clarify how we performed our
radiocarbon age calibration.

For radiocarbon calibration and age model calculation, we used PaleoDataView (PDV) (Langner and
Mulitza, 2019), a software that implements IntCal20 (Reimer et al., 2020) for the initial calibration of ¹⁴C
ages. However, we further corrected these IntCal20-calibrated ¹⁴C ages using location- and time-specific
reservoir age estimates from Butzin et al. (2017). To produce the age models, we used the Bacon
algorithm (Blaauw and Christen, 2011) within PDV with its respective default parameters.

To clarify this approach, we have now expanded the explanation about this correction in the revised
manuscript (lines 437-465). These reservoir age estimates were derived from the Hamburg Large-Scale
Geostrophic ocean general circulation model, which simulated the spatial and temporal variability of
marine radiocarbon ages in surface waters over the last 50,000 years. This 3D ocean modeling approach
enables a robust marine reservoir age correction, particularly in low- to mid-latitude regions during the
Holocene. However, uncertainties increase further back in time and at higher latitudes, such as the Arctic
Ocean (Butzin et al., 2017). While we consider this approach valid for the spatial and temporal scales of
this study, we acknowledge that some critical voices remain (Heaton et al., 2022; Heaton et al., 2023).

To demonstrate that our radiocarbon calibration approach is effective and produces results comparable
to a calibration based on Marine20 (Heaton et al., 2020), we have re-calibrated all ¹⁴C ages of this study
using Marine20. For this recalibration, we used the 8.2HTML version of CALIB, the radiocarbon
calibration tool developed by M. Stuiver, P.J. Reimer, and R. Reimer (<http://calib.org/calib/>). Additionally,
we applied a marine reservoir age correction to the Marine20-calibrated radiocarbon ages by calculating
the mean \$\Delta R\$ and mean \$\Delta R_{\text{error}}\$ from the ten closest locations to each core site (ODP 1063, ODP 1059, and
GeoB18529-2). The Marine20-calibrated radiocarbon results are presented here in Response Table 1,
while the mean \$\Delta R\$ values and their respective errors are provided in Response Table 2.

We found that the differences between the radiocarbon calibration approach used in this study and the

uncorrected Marine20-based approach result in an average absolute deviation of approximately 172
 89 years (6.2%). However, when comparing our calibrated ages with the marine reservoir age-corrected
 Marine20-calibrated ages (bold columns in Response Table 1), the mean absolute difference is reduced
 to ~96 years (2.7%) between both approaches.

Since the corrected Marine20 results fall within the 95% uncertainty range of our approach, and we are
 working with a millennial-scale temporal resolution, we consider the calibration method used in this
 study to be robust, particularly for mid-latitude sediments cores (Response Table 2) of Holocene age.
 However, we acknowledge that the differences of the results between both approaches might be more
 pronounced and significant when working with (I) temporal resolutions of centennial time-scales or
 higher, (II) cores located beyond the low- to mid-latitudes, particularly in the Arctic Ocean, and (III)
 periods predating the Holocene (Butzin et al., 2017; Heaton et al., 2022; Heaton et al., 2023).

 We agree with Reviewer #4 that the previous explanations about the ¹⁴C age calibration and the age
 model calculation were not detailed enough and therefore leading to confusion. Based on the feedback
 from Reviewer #4, we decided to rework the corresponding section of the manuscript by expanding on
 our ¹⁴C calibration approach (lines 437-465).

For the revised chronology of Core ODP 1063, we used the published radiocarbon ages of Lippold et al.
 (2019) together with five new radiocarbon ages, calibrated all of them and produced the age model using
 the approach described above. We also added this information to the revised manuscript, lines 444-452
 and 454-465.

We additionally added a chapter to the supplement called “Age Models”. In this chapter we provide the
 respective age-depth profiles and expand on our explanation (Supplement lines 64-163).

 **Response Table 1: Uncalibrated ¹⁴C ages compared with IntCal20 (Reimer et al., 2020) calibrated ¹⁴C ages**
 **(including reservoir ages from Butzin et al. (2017)) and Marine20 (Heaton et al., 2020) calibrated ¹⁴C ages without**
 **and with a mean ΔR. The IntCal20 ages were calibrated using the PaleoDataView software (Langner and Mulitza,**
 **2019), while the Marine20 ¹⁴C ages were calibrated using the CALIB 8.2HTML Radiocarbon Calibration software**
 **(<http://calib.org/calib/>). Since this study focuses on millennial time-scales and for the sake of simplicity, we have**
 **rounded all ages to two decimal places.**

Core	Lab ID	Age dated [ka BP]	Age dated sd [± ka]	IntCal20 calib. age [weighted mean; ka BP]	IntCal20 calib. age min. [95%, ka BP]	IntCal20 calib. age max. [95%, ka BP]	Marine20 calib. Age [ka BP]	Marine20 calib. age with mean ΔR [ka BP]
GeoB18529-2	418847	1.25	0.03	0.76	0.67	0.91	0.645	0.70
GeoB18529-2	418848	4.51	0.03	4.66	4.45	4.83	4.505	4.57
GeoB18529-2	BE-12549.1.1	8.70	0.05	9.25	9.03	9.44	9.176	9.23
GeoB18529-2	BE-12550.1.1	10.24	0.06	10.94	10.70	11.18	11.202	11.27
GeoB18529-2	BE-12551.1.1	13.20	0.05	14.66	14.09	15.22	15.041	15.10
GeoB18529-2	BE-12552.1.1	14.38	0.05	16.46	15.96	16.99	16.577	16.64
GeoB18529-2	418849	16.52	0.05	18.96	18.65	19.40	19.030	19.09
ODP1059	BE-19464.av	3.61	0.23	3.55	2.94	4.22	3.345	3.63
ODP1059	BE-19462.1.1	5.28	0.03	5.66	5.48	5.89	5.453	5.69
ODP1059	BE-16592.1.1	2.98	0.04	2.79	2.54	2.94	2.592	2.86

ODP1059	BE-16593.1.1	3.42	0.05	3.28	3.08	3.45	3.118	3.38
ODP1059	BE-16594.1.1	3.86	0.04	3.82	3.64	4.06	3.644	3.93
ODP1059	BE-19466.1.1	4.82	0.03	5.13	4.88	5.31	4.897	5.16
ODP1059	BE-19463.av	7.81	0.07	8.24	8.04	8.39	8.094	8.33
ODP1059	BE-19465.1.1	12.86	0.15	14.64	14.10	15.16	14.492	14.82
ODP 1063	74957.1.1	1.27	0.04	0.82	0.69	0.93	0.668	0.76
ODP 1063	74958.1.1	2.22	0.05	1.82	1.63	1.99	1.639	1.75
ODP 1063	81619.1.1	2.77	0.06	2.55	2.35	2.73	2.316	2.44
ODP 1063	72008.1.1	1.05	0.05	0.64	0.55	0.73	0.494	0.56
ODP 1063	81620.1.1	1.20	0.06	0.78	0.67	0.91	0.602	0.69
ODP 1063	72009.1.1	3.12	0.05	2.91	2.76	3.14	2.755	2.86
ODP 1063	BE-16595.1.1	3.69	0.04	3.61	3.45	3.82	3.435	3.55
ODP 1063	81621.1.1	3.73	0.07	3.66	3.45	3.88	3.490	3.60
ODP 1063	BE-16597.1.1	4.22	0.05	4.29	4.09	4.51	4.130	4.25
ODP 1063	BE-16598.1.1	4.88	0.04	5.10	4.88	5.30	4.970	5.09
ODP 1063	72010.1.1	5.67	0.06	6.09	5.92	6.28	5.865	5.97
ODP 1063	85099.1.1	6.67	0.07	7.13	6.91	7.41	6.974	7.06
ODP 1063	85100.1.1	6.93	0.07	7.42	7.27	7.57	7.243	7.32
ODP 1063	85101.1.1	8.16	0.07	8.70	8.45	8.98	8.466	8.59
ODP 1063	85102.1.1	8.72	0.08	9.34	9.04	9.53	9.203	9.30
ODP 1063	85103.1.1	9.89	0.07	10.92	10.60	11.18	10.732	10.84
ODP 1063	72012.1.1	10.00	0.09	10.95	10.60	11.22	10.876	10.98
ODP 1063	BE-16602.1.1	10.95	0.05	12.43	12.06	12.68	12.273	12.37

117
118
119
120

Response Table 2: Mean ΔR used for the Marine20 reservoir age correction. The mean ΔR was calculated via the Marine Reservoir Correction Database feature of CALIB 8.2HTML Radiocarbon calibration software (<http://calib.org/calib/>).

Core	Core latitude [°N]	Core longitude [°E]	Number of considered points	Weighted mean ΔR	Mean ΔR_{error} [σ]
GeoB18529-2	41.876833	-47.557167	10	-49	66
ODP1059	31.67422	-75.41878	10	-186	143
ODP 1063	33.68620	-57.61512	10	-87	99

R4.2

I presume that the dates are being archived on pangaea, but this isn't explicitly stated (would be helpful
to see supplementary data sheet which apparently lists the data).

We regret not mentioning in the previous version of the manuscript that these data were already
available in the co-submitted Excel Data Sheet, also containing the U, Th and Pa raw data. However,
based on this suggestion from Reviewer #4, we have added an additional table with the ¹⁴C ages to the
Supplement (Supplementary Table 2), that provides all information about the radiocarbon dates. In the
revised age model section of the manuscript, we now refer to the new ¹⁴C Supplement table (lines 290,
440, 452, 460). Additional information to the radiocarbon data can be found in the Excel Data Sheet. We
now also refer to it in lines 445, 448-449.

Regarding PANGAEA and the Excel Data Sheet, we have previously stated in the Data Availability section
of the manuscript that the data from this study will be made available in PANGAEA and included a
placeholder link.

However, the three datasets for (I) radionuclides, (II) ¹⁴C ages, and (III) GAM were recently accepted by
PANGAEA. The data are currently under moratorium and are password-protected. We have included
reviewer access links in the manuscript to allow for data inspection (lines 604-607). This link is active until
the fourth of October this year. We further refer to the Excel Data Sheet included in the first *Nature*
*Communications* submission.

R4.3
Extra dates are also included for ODP 1059. Again insufficient information is given to evaluate the
chronology.

Based on these comments we decided to rework the age model section of the manuscript as explained
above (lines 437-469).

We have now added a supplementary table that provides a transparent overview of all considered
radiocarbon ages, both calibrated and uncalibrated, along with their respective uncertainties
(Supplementary Table 2). Additionally, in the revised manuscript, we refer to this table in the age model
section (lines 290, 440, 452, 460), as well as to the corresponding Excel Data Sheet (lines 445, 448-449),
which contains these data along with additional information on the radiocarbon data (e.g. dated
material).

We provide the respective age-depth profiles in the newly added Supplement chapter "Age Models" and
expand on our explanation (Supplement lines 119-155).

R4.4
A new age-depth model is made for GeoB18529-2. No information is given for the bacon settings. I
presume the default priors and other settings are used. This is probably reasonable for this high-
sedimentation site, and not reporting this information is not unusual. Again, I don't understand why
intcal20 is being used on marine dates. The age-depth model for this core is shown. It is not the happiest
chronology I have seen: two of the seven dates are completely outside model's 95% uncertainty. It is
possible that the sedimentation rate is more varied than the current model suggests. Only more dates
will improve the model. Fortunately, because the reconstructed AMOC is not very dynamic, the demands
on the chronology are not too great.

We regret the misunderstanding of the radiocarbon age calibration. We hope to clarify any concerns with
our radiocarbon calibration with the answer provided in R4.1.

As stated above we updated the age model section in the revised manuscript and included the statement
that we only used the default PDV Bacon settings and Bacon version 2.3.9.1 (lines 459-465).

We acknowledge that sedimentation rates may be more variable than our current model suggests.
However, as noted by Reviewer #4, the revisions made in the first review round (i.e., correcting the
temporal resolution from the previously stated centennial-scale to the now revised temporal scale)
indicate that such uncertainties are within the expected error range for the millennial timescale.
Since this age model was developed entirely from new data, without refinement based on previous
studies (e.g., ODP 1059 and ODP 1063), we have included the corresponding age-depth profile
(Supplementary Fig. 5). While Reviewer #4 correctly points out that two of the seven calibrated
radiocarbon ages fall outside the uncertainty range, we would like to clarify that the uncertainty
envelope provided for the calibrated radiocarbon ages represents the 95% confidence range (see the
updated Supplementary Table 2), whereas the uncertainty reported for the Bacon model follows the
97.5% confidence envelope. Accordingly, we have corrected the description of Supplementary Fig. 5 to
reflect this, adjusting the stated Bacon uncertainty from 95% to 97.5%.

R4.5

line 194 "partly very different" rephrase - perhaps "occasionally very different"
Changed accordingly (line 194).

R4.6

Fig 4 (with implications for the other figures). I find the red and orange quite difficult to distinguish.
Please try to increase the contrast between colours and consider using shape to help distinguish the
points.

We thank Reviewer #4 for this good suggestion. We have added different symbol shapes to distinguish
between the data points of the respective records.

In this context we additionally changed the symbol shapes of Supplementary Fig. 4.

R4.7

line 347 "6.5 ka BP on until the PI" delete "on"
Changed accordingly (line 348).

R4.8

line 468 "for ^{231}Pa below 2 %" -> and below 2% for ^{231}Pa
Changed accordingly (lines 483-485).

R4.9

Fig S3. The time axis on this plot runs from left to right, whereas on the other plots, time runs right to
left. The scale for the grey line in the upper panels is not explicitly given.

For Supplementary Fig. 3: We have inverted the x-axis and changed the tick division of the y-axis to show
more frequently used labels, as requested.

Additionally, we inverted the x-axis of Supplementary Fig. 13 (previously Supplementary Fig. 7), to fit to
the changes made to Supplementary Fig. 3.

R4.10

I don't fully understand the detrital correction. Is it possible that the increased influx of detritus at 4.2 ka
in ODP 1063 is biasing the Pa/Th record?

The detrital correction corrects the excess of ^{231}Pa and ^{230}Th in the bulk concentration for lithogenic
contributions of ^{231}Pa and ^{230}Th . For this it is assumed a $(^{238}\text{U}/^{232}\text{Th})_{\text{det}}$ in detrital material, which is in secular
equilibrium $^{238}\text{U}_{\text{det}} = ^{230}\text{Th}_{\text{det}}$. Here we assume $(^{238}\text{U}/^{232}\text{Th})_{\text{det}} = 0.47$ (Bourne et al., 2012; Francois et al.,
2004).

Reviewer #4 is right, that in theory a highly variable input of detrital material may change the lithogenic
 background and as a consequence may alter the resulting $^{231}\text{Pa}/^{230}\text{Th}$. However, a change in $(^{238}\text{U}/^{232}\text{Th})_{\text{det}}$
 would affect both ^{231}Pa and ^{230}Th in almost the same way. In particular for young Holocene $^{231}\text{Pa}/^{230}\text{Th}$,
 with the excess fractions representing the by far major fractions of ^{231}Pa and ^{230}Th , changes in
 $(^{238}\text{U}/^{232}\text{Th})_{\text{det}}$ are not expected to have a big impact.
 To demonstrate this, we attached here the $^{231}\text{Pa}/^{230}\text{Th}$ of ODP 1063 with different values of $(^{238}\text{U}/^{232}\text{Th})_{\text{det}}$
 (Response Fig. 1b). Even with unrealistic high and low values of 0.3 and 0.7, $(^{238}\text{U}/^{232}\text{Th})_{\text{det}}$ of ODP 1063
 varies between 0.47 and 0.6 during the Holocene, with an average of 0.53 (Response Fig. 1a)), respectively,
 the $^{231}\text{Pa}/^{230}\text{Th}$ does not change significantly. The peak in $^{231}\text{Pa}/^{230}\text{Th}$ around 4.2 ka BP is thus not a result
 of a change of the lithogenic background, but must be related to the increase of ^{231}Pa -scavenging due to
 increased detrital particle flux as monitored by the ^{232}Th -fluxes in Fig. 5.

**Response Figure 1: $^{231}\text{Pa}/^{230}\text{Th}$ of core ODP 1063 with varying detrital correction. a) $(^{238}\text{U}/^{232}\text{Th})_{\text{det}}$ of ODP 1063 over**
 **the Holocene. b) Black line gives the $^{231}\text{Pa}/^{230}\text{Th}$ calculation for a detritus factor of 0.47 (as assumed in the manuscript).**
 **The gray area comprises the range of $^{231}\text{Pa}/^{230}\text{Th}$ for a detritus factor of 0.7 (upper limit) to 0.3 (lower limit).**

**References**

Blaauw, M., and Christen, J. A., 2011, Flexible paleoclimate age-depth models using an autoregressive

gamma process: *Bayesian Analysis*, v. 6, no. 3.

Bourne, M. D., Thomas, A. L., Mac Niocaill, C., and Henderson, G. M., 2012, Improved determination of

marine sedimentation rates using ^{230}Th s: *Geochemistry, Geophysics, Geosystems*, v. 13, no. 9.

Butzin, M., Köhler, P., and Lohmann, G., 2017, Marine radiocarbon reservoir age simulations for the past

50,000 years: *Geophysical Research Letters*, v. 44, no. 16, p. 8473-8480.

Francois, R., Frank, M., Rutgers van der Loeff, M. M., and Bacon, M. P., 2004, ^{230}Th normalization: An

essential tool for interpreting sedimentary fluxes during the late Quaternary: *Paleoceanography*,

v. 19, no. 1.

Heaton, T. J., Bard, E., Bronk Ramsey, C., Butzin, M., Hatté, C., Hughen, K. A., Köhler, P., and Reimer, P. J.,

2022, A Response to Community Questions on the Marine20 Radiocarbon Age Calibration Curve:

Marine Reservoir Ages and the Calibration of ^{14}C Samples from the Oceans: *Radiocarbon*, v. 65,

no. 1, p. 247-273.

Heaton, T. J., Butzin, M., Bard, E., Bronk Ramsey, C., Hughen, K. A., Köhler, P., and Reimer, P. J., 2023,

Marine Radiocarbon Calibration in Polar Regions: A Simple Approximate Approach Using

Marine20: *Radiocarbon*, v. 65, no. 4, p. 848-875.

Heaton, T. J., Köhler, P., Butzin, M., Bard, E., Reimer, R. W., Austin, W. E. N., Bronk Ramsey, C., Grootes, P.

254 M., Hughen, K. A., Kromer, B., Reimer, P. J., Adkins, J., Burke, A., Cook, M. S., Olsen, J., and Skinner,

255 L. C., 2020, Marine20—The Marine Radiocarbon Age Calibration Curve (0–55,000 cal BP):

*Radiocarbon*, v. 62, no. 4, p. 779-820.

Langner, M., and Mulitza, S., 2019, Technical note: PaleoDataView – a software toolbox for the collection,

homogenization and visualization of marine proxy data: *Climate of the Past*, v. 15, no. 6, p. 2067-

2072.

Lippold, J., Pöppelmeier, F., Süfke, F., Gutjahr, M., Goepfert, T. J., Blaser, P., Friedrich, O., Link, J. M.,

Wacker, L., Rheinberger, S., and Jaccard, S. L., 2019, Constraining the variability of the Atlantic

Meridional Overturning Circulation during the Holocene: *Geophysical Research Letters*, v. 46, no.

20, p. 11338-11346.

Reimer, P. J., Austin, W. E. N., Bard, E., Bayliss, A., Blackwell, P. G., Bronk Ramsey, C., Butzin, M., Cheng, H.,

Edwards, R. L., Friedrich, M., Grootes, P. M., Guilderson, T. P., Hajdas, I., Heaton, T. J., Hogg, A. G.,

Hughen, K. A., Kromer, B., Manning, S. W., Muscheler, R., Palmer, J. G., Pearson, C., van der Plicht,

267 J., Reimer, R. W., Richards, D. A., Scott, E. M., Southon, J. R., Turney, C. S. M., Wacker, L., Adolphi,

F., Büntgen, U., Capano, M., Fahrni, S. M., Fogtmann-Schulz, A., Friedrich, R., Köhler, P., Kudsk, S.,

Miyake, F., Olsen, J., Reinig, F., Sakamoto, M., Sookdeo, A., and Talamo, S., 2020, The IntCal20

Northern Hemisphere Radiocarbon Age Calibration Curve (0–55 cal kBP): *Radiocarbon*, v. 62, no.

4, p. 725-757.

**Author's response to reviews on the manuscript "Low variability of the Atlantic Meridional**
**Overturning Circulation throughout the Holocene" (NCOMMS-24-61663B)**

We sincerely thank Reviewer #4 and the Editor for their time and effort reviewing our manuscript.

We acknowledge that our previous response did not fully address the concerns regarding our age models.
While we believe that our previous approach was robust, we recognize Reviewer #4's concerns and the
request to adopt a Marine20-based calibration approach.

In this response letter, we demonstrate that we have addressed this request by re-calibrating and re-
calculating all new age models shown in this study, using the Marine20 calibration curve and rbacon
modelling tool. Importantly, regardless of the calibration approach used, the findings of this study remain
consistent. Still, we updated all figures containing age-related information, revised all relevant numerical
values in both the manuscript and the supplement, and made minor adjustments to the text to
accommodate the revised results. We hope that these revisions resolve any remaining age model related
concerns.

In the following, we present the Reviewers' comments in black, with our responses provided in blue. Stated
line numbers refer to the revised versions of the manuscript and the supplement without "tracked
changes". Additionally, we are also providing a "tracked changes" version of both documents, so that it is
possible to follow in detail each individual change.

We hope that our revisions and clarifications will fulfill the Editors' as well as the Reviewers' requirements
and expectations.

**Reviewer #4**

Review of "Low variability of the Atlantic Meridional Overturning Circulation throughout the Holocene"
Gerber et al

The revisions since the previous version have greatly improved the presentation of the chronology.
However, I am still not convinced by the approach taken to calibrate the radiocarbon dates, but concede
that the differences are probably minor.

The standard way to calibrate marine radiocarbon dates is to use the marine calibration curve, derived
from the atmospheric calibration curve using a highly simplified carbon model, and a delta R value to
account for the deviation between the modelled reservoir effect and the local conditions. The challenge
is that delta R varies in time and space and for many areas it is poorly constrained.

An alternative would be to use a more complex ocean model forced by palaeoclimate to generate a
bespoke calibration curve for each core location. The challenge is that the model is much slower and so
cannot be run many times to explore its uncertainty.

With both these approaches, the resulting calibration curve is smoothed compared with the atmospheric
curve.

I understand that for this manuscript, the reservoir age estimated with a relatively complex earth system
model is being used with the atmospheric calibration curve. I don't think this is appropriate as there is no
smoothing of the calibration curve that is inherent in the real ocean and in the carbon models. This can

possibly be seen in the some of the calibrated dates that have complex PDFs, which you don't normally
see for marine dates given the smoothness of the marine calibration curve.

Assuming a bespoke calibration curve with uncertainties cannot be generated, a more appropriate
method would be to use the delta-R from the complex model with marine20. I'm not quite sure how the
delta-R would be defined from the model (perhaps against the modelled global mean).

As the authors show in the response to reviewers, the choice of methods does not have a large effect,
but I think it is important to use the most reliable methods.

We thank Reviewer #4 for this feedback.

As summarized above by Reviewer #4, both our original approach and a Marine20-based calibration have
advantages and limitations. As an increasing number of studies have adopted the atmospheric calibration
curve in combination with location- and time-specific marine reservoir age estimates, we previously used
this approach. However, Reviewer #4 is correct in pointing out that, while this method is becoming more
common [1-10], it is not the standard procedure for calibrating radiocarbon ages of marine samples. We
therefore acknowledge the Reviewer's concerns regarding our initial approach and have now adopted the
requested Marine20-based radiocarbon calibration. Below, we provide a brief summary of the method
used for generating the revised age models. We also briefly describe the resulting changes.

We first re-calibrated all radiocarbon ages (Supplementary Table 2) using the rbacon tool (v.3.3.1) [11]
with the Marine20 calibration curve [12], applying the weighted mean \$\Delta R\$ value derived from the ten
geographically closest marine reservoir offsets listed in the Marine20 database [13]. We then re-calculated
the Bacon age models (the used parameters were added to Supplementary Table 3). To align with this new
age model approach, we revised the age model section of the manuscript (lines 460-468). Additionally, we
updated the chronology section in the Supplement by incorporating the new age-depth figures
(Supplementary Figs. 5, 6, 7, 9, 10), revising their respective captions and adjusting the description of our
chronologies accordingly (Supplementary lines: 92-97, 105-108, 118-120, 132, 134, 135, 147, 148, 150).

In this context we also updated all Figures containing time information (Figs. 2, 4, 5 and Supplementary
Figs. 1, 2, 4) and Table 1. We re-calculated the Generalized Additive Model (GAM) based on the revised
age models and observed only minor differences compared to the GAM presented in the previous version
of the manuscript. While the overall structure of the GAM remains unchanged, the excursions are now
slightly more pronounced. To reflect these revised results, we have made slight updates to the manuscript
(lines 28-31, 134, 136-137, 250-253, 274, 283, 284-287, 289, 291, 295, 354, 385-386, 387, 391-393, 567,
574) and the respective figure descriptions (lines 149, 243-244, 350-351). Additionally, we are updating
the PANGAEA data sheets.

We believe that these changes have further improved the manuscript. The high consistency of our findings,
regardless of the radiocarbon calibration approach, should help dispel any remaining concerns.

Minor points

The abbreviation mbsf does not seem to be defined.

This abbreviation is now defined (Supplement line 79).

The SI (and the response to reviewers) describes a 97.5% confidence interval. This is an unusual interval
to choose, and differs from the default 95% interval (at least in rBacon). The 95% interval spans from
2.5% to 97.5%. Please check the interval has been reported correctly.

Since we generated new age models, we decided to display the 2σ (95% probability) intervals in the age-
depth figures. This information has been updated in the revised captions of all age-depth figures
(Supplementary lines 88, 102, 114, 142, 157) and is also included in the newly added Supplementary Table
3, under the “prob” argument of the rbacon parameters.

Table S2 "Age dated" and "Age dated sd" are strangely worded. Perhaps "14C age". The calibrate age
range could be presented in one column rather than two.

Supplementary Table 2 has been revised to show the newly calibrated radiocarbon ages. We now state
“¹⁴C age” and “¹⁴C age sd [±]”.

The calibrated upper and lower 95% age intervals differ from the median calibrated age. We thus believe
it is most appropriate to present them in two separate columns. However, we agree with Reviewer #4 that
the table is quite dense, but given that it appears in the Supplement, we consider the revised format to be
acceptable.

Fig S10. Something is strange with the first radiocarbon data - the plotted PDF is a triangle that does not
include the weighted mean and does not correspond to the uncertainty whiskers. Please check this
figure has rendered correctly.

Since we generated a completely new age model for ODP 1059 A, the age-depth figure has changed, and
the previous error has now been corrected (Supplementary Fig. 10).

**References**

- 1. Max, L., et al., *Subsurface ocean warming preceded Heinrich Events*. Nat Commun, 2022. **13**(1):
p. 4217.
- 2. Muglia, J., et al., *A global synthesis of high-resolution stable isotope data from benthic*
*foraminifera of the last deglaciation*. Scientific Data, 2023. **10**(1): p. 131.
- 3. Jonkers, L., et al., *Integrating palaeoclimate time series with rich metadata for uncertainty*
*modelling: strategy and documentation of the PalMod 130k marine palaeoclimate data*
*synthesis*. Earth System Science Data, 2020. **12**(2): p. 1053-1081.
- 4. Köhler, P. and S. Mulitza, *No detectable influence of the carbonate ion effect on changes in stable*
*carbon isotope ratios ($\delta^{13}C$) of shallow dwelling planktic foraminifera over the past 160 kyr*.
*Climate of the Past*, 2024. **20**(4): p. 991-1015.
- 5. Campos, M.C., et al., *Constraining Millennial-Scale Changes in Northern Component Water*
*Ventilation in the Western Tropical South Atlantic*. *Paleoceanography and Paleoclimatology*,
2020. **35**(7).
- 6. Ferreira, J.Q., et al., *Changes in obliquity drive tree cover shifts in eastern tropical South America*.
*Quaternary Science Reviews*, 2022. **279**.
- 7. Saini, J., et al., *Holocene variability in sea-ice conditions in the eastern Baffin Bay-Labrador Sea –*
*A north–south biomarker transect study*. *Boreas*, 2022. **51**(3): p. 553-572.
- 8. García Chaporí, N., et al., *Holocene palaeoceanographic history of the western South Atlantic*.
*Journal of South American Earth Sciences*, 2022. **117**.
- 9. Lešić, N.-M., et al., *Glacimarine sediments from outer Drygalski Trough, sub-Antarctic South*
*Georgia – evidence for extensive glaciation during the Last Glacial Maximum*. *Quaternary Science*
*Reviews*, 2022. **292**.
- 10. Martins, A.K., et al., *Links between precipitation patterns over eastern tropical South America*
*and productivity in the western tropical South Atlantic Ocean during the last deglacial*.
*Quaternary International*, 2023. **667**: p. 29-40.
- 11. Blaauw, M. and J.A. Christen, *Flexible paleoclimate age-depth models using an autoregressive*
*gamma process*. *Bayesian Analysis*, 2011. **6**(3).
- 12. Heaton, T.J., et al., *Marine20—The Marine Radiocarbon Age Calibration Curve (0–55,000 cal BP)*.
*Radiocarbon*, 2020. **62**(4): p. 779-820.
- 13. Reimer, P.J. and R.W. Reimer, *A Marine Reservoir Correction Database and On-Line Interface*.
*Radiocarbon*, 2016. **43**(2A): p. 461-463.

General comments

This paper is proposing a new reconstruction of the AMOC over the Holocene, based on 5 Pa/Th proxy records in the Atlantic Ocean and scaled towards AMOC reconstruction using Bern3D model. The reconstruction obtained is showing quite few variations over the Holocene. It is then highlighted that given the density of proxy records and their time resolution, it is not possible to detect AMOC weakening larger than about 5 Sv on a time frame shorter to about 200 years.

This paper is based on a robust approach and useful proxy records for reconstruction of the AMOC. In this respect this is a nice paper. However, the statements from the title and some others throughout the paper are just disagreeing with the evidences provided in the paper.

For instance, Fig. 3, which is the climax of the paper in my view, is just demonstrating that the approach cannot allow to reconstruct centennial variability, except for time scale larger than 200 years and AMOC changes of very substantial amplitude (about more than 30% of the present-day amplitude!). Thus, while this reconstruction is of interest, it cannot, by construction, supports the claims of the paper.

What this reconstruction can solve has more to do with millennial variability. To solve centennial variability, you actually need to have proxy records with a time resolution about one order of magnitude lower than the period you are analysing. This can be simply understood in the scheme provided in this review (Fig. R1). To solve a 100-yr cycle, you need to be able to sample the high phase and low phase of the cycle, which are separated by about 50 years. To have robust results, I think (and this can be proven properly, see specific comments) you might need resolution of about 10 years, which is far from the resolution of proxy records used. Due to this editorial flaw (why discussing centennial variability while your records do not really allow to access it?), I cannot support the publication of this paper in its present form.

Fig. R1: Scheme representing a crude first estimate of the number of points that are requested to be able to resolve centennial variability from reconstruction. The blue line represents a variation of a time series with a period of 100 years. The blue points are the minimum and maximum of the cycle. The red points represent where additional sample data might be necessary to be able to assess from reconstruction the centennial variability.

Specific comments

- Line 27: provide an estimate of the time resolution
- Line 28-29: this sentence is unclear
- Line 57-58: This sentence is wrong, there exists (at least) one quantitative estimate of the AMOC over a large part of the Holocene in Jomelli et al. (2022).
- Line 60: “fundamental principle” is a bit strong. This is an assumption and has nothing to do with a fundamental physical principle. This proxy, as much of others, is still debated in terms of exact representativity.
- Line 71-72: Is there any instrumental validation of this proxy?
- Line 108: “Higher than 0.5 ka” is not very precise. Lower than?
- Line 123: “centennial scale resolution” is not allowing to assess centennial variability (cf. Fig. R1)
- Line 125-126: it might be nice to use pseudo-proxy approach to evaluate the representativity of the sampling in terms of AMOC reconstruction (e.g. Ayache et al. 2018). This means using a model modelling this proxy and see if the sampling is permitting to correctly represents the AMOC. It is unclear if this has been done in former publications. I doubt so for this specific sampling. This is a necessary validation test to have any confidence in this reconstruction.

- Line 159: a pseudo-proxy approach might be also a way to evaluate the type of variability you are able to reproduce from your temporal sampling. Which temporal sampling in a model do you need to be able to reproduce centennial variability from a given model (e.g. Jiang et al. 2021)
- Line 179-180: on Fig. 4, one can clearly see that the reconstruction is far flatter than the original proxy records, which were showing some substantial variability, which is totally smoothed by the model and the reconstruction procedure.
- Line 193-194: what is the internal variability of this model? does it have any multi-centennial variability when run in preindustrial conditions? If not, it is very likely that it might not be able to reconstruct centennial variability, which can be found in a number of CMIP6 models (e.g. Bonnet et al. 2021). Also what are the biases for present-day water masses representation? This might strongly affect the capability of reconstruction the AMOC in the past based on depth of water masses. For instance, it is already difficult to reconstruct the AMOC in data assimilation system before the Argo period, because the observation sampling and the biases in the model are too strong, cf. Karspeck et al. (2015)
- Line 221: “modest (-4,9 Sv) AMOC weakening”. I do not think such a weakening over 100 years is modest! It is larger than what is projected in some CMIP6 models for 2100 and represents more than 25% of present-day AMOC! As a point of comparison, Jomelli et al. (2022) estimated that mid-Holocene might be possibly 3-4 Sv larger than present-day. Thus, the method and proxy records used in this paper does not allow to contradict results from this former AMOC reconstruction. What this approach is allowing to conclude is just that there were no DO-like events during the Holocene, which is already well known!
- Line 278: It is useful to keep in mind that response of climate models to a given amount of freshwater is very model dependent (cf. Stouffer et al. 2006, and still true in most recent models, cf. Jackson et al. 2023).
- Line 302: Your reconstruction is not allowing to exclude the occurrence of a very short-scale events of few decades, related for instance only to a subpolar gyre instability (e.g. Sgubin et al. 2019).
- Line 315-320: In the higher resolution AMOC reconstruction from Ayache et al. (2018), there is an AMOC fluctuation around 4.2 ka. Can it be possible that it is not detected here due to low sensitivity to AMOC changes of the method used?
- Line 348: How is the uncertainty of 1.1 Sv cited in parenthesis is computed? It seems largely underestimated in my view. Same remark with the number 1.3 Sv line 180.

Bibliography

Bonnet R., et al. (2021) Increased risk of near term global warming level due to a recent AMOC weakening. *Nature Communications*, 12, Article number: 6108.

Jackson L., et al. (2023) Understanding AMOC stability: the North Atlantic Hosing Model Intercomparison Project. *Geoscientific Model Development*, 16, 1975-1995.

- Jiang, W., Gastineau, G. & Codron, F. (2021) Multicentennial variability driven by salinity exchanges between the atlantic and the arctic ocean in a coupled climate model. *J. Adv. Model. Earth Syst.* 13, e2020MS002366.
- Jomelli V et al. (2022) AMOC control on millennial-scale glacier changes during the Holocene. *Nature Communications*, 13(1), 1419.
- Karspeck et al. (2015) Comparison of the Atlantic meridional overturning circulation between 1960 and 2007 in six ocean reanalysis products. *Clim Dyn*
DOI 10.1007/s00382-015-2787-7
- Sgubin G. , et al. (2019) The Impact of Possible Decadal-Scale Cold Waves on Viticulture over Europe in a Context of Global Warming. *Agronomy*, 9, 397;
doi:10.3390/agronomy9070397
- Stouffer R et al (2006) Investigating the causes of the response of the thermohaline circulation to past and future climate changes. *J Climate* 19:1365–1387